



# Rapid fragmentation of Thwaites Eastern Ice Shelf, West Antarctica

Douglas I. Benn[1], Adrian Luckman[2], Jan A. Åström[3], Anna J. Crawford[1], Stephen L. Cornford[2], Suzanne Bevan[2], Rupert Gladstone[4], Thomas Zwinger[3], Karen Alley[5], Erin Pettit[6] and Jeremy Bassis[7]

[1]School of Geography and Sustainable Development, University of St Andrews, St Andrews, KY16 9AL, UK
[2]Department of Geography, Swansea University, Swansea, SA2 8PP, UK
[3]CSC-IT Center for Science, FI-02101 Espoo, Finland
[4]The Arctic Centre, University of Lapland, 96101 Rovaniemi, Finland
[5]Department of Environment and Geography, University of Manitoba, Winnipeg, MB R3T 2M6, Canada
[6]College of Earth, Ocean and Atmospheric Sciences, Oregon State University, Corvallis, OR 97331-5503, USA
[7]Department of Earth and Environmental Sciences, University of Michigan, Ann Arbor, MI 48109-1005, USA

*Correspondence to:* Doug Benn (dib2@st-andrews.ac.uk)

**Abstract** Ice shelves play a key role in the dynamics of marine ice sheets, by buttressing grounded ice and limiting rates of ice flux to the oceans. In response to recent climatic and oceanic change, ice shelves fringing the West Antarctic Ice Sheet (WAIS) have begun to fragment and retreat, with major implications for ice sheet stability. Here, we focus on the Thwaites Eastern Ice Shelf (TEIS), the remaining pinned floating extension of Thwaites Glacier. We show that TEIS has undergone a process of fragmentation in the last five years, including brittle failure along a major shear zone, formation of tensile cracks on the main body of the shelf, and release of tabular bergs on both eastern and western flanks. Simulations with the Helsinki Discrete Element Model (HiDEM) show that this pattern of failure is associated with high backstress from a submarine pinning point at the distal edge of the shelf. We show that a significant zone of shear upstream of the main pinning point developed in response to the rapid acceleration of the shelf between 2002 and 2006, seeding damage on the shelf. Subsequently, basal melting and positive feedbacks between damage and strain rates weakened TEIS, allowing damage to accumulate. Thus, although backstress on TEIS has likely diminished through time as the pinning point has shrunk, accumulation of damage has ensured that the ice in the shear zone has remained the weakest link in the system. Experiments with the BISICLES ice sheet model indicate that additional damage to or unpinning of TEIS are unlikely to trigger significantly increased ice loss from WAIS, but the calving response to loss of TEIS remains highly uncertain. It is widely recognised that ice-shelf fragmentation and collapse can be triggered by hydrofracturing and/or unpinning from ice shelf margins or grounding points. Our results indicate a third mechanism, *backstress-triggered failure*, that can occur when ice ffractures in response to stresses associated with pinning points. In most circumstances, pinning points are essential for ice shelf stability, but as ice shelves thin and weaken the concentration of backstress in damaged ice upstream of a pinning point may provide the seeds of their demise.





## 1. Introduction

Ice shelves play a key role in the dynamics of marine ice sheets. By transmitting resistive stresses from lateral or basal pinning points to the grounding line, ice shelves buttress grounded portions of the ice sheet and constrain ice flow (e.g. Doake et al., 1998; DuPont and Alley, 2005). If buttressing is reduced or lost following the retreat or disintegration of ice shelves, tributary glaciers can accelerate and increase ice flux to the ocean (Scambos et al., 2004). Accelerated ice discharge may be temporary and reversible, but modelling studies indicate that in some circumstances (e.g. where the bed is steeply retrograde) ice-shelf break-up may initiate sustained loss of grounded ice and irreversible retreat through the marine ice sheet and marine ice cliff instabilities (Schoof, 2012; Sun et al., 2020; DeConto et al., 2021; Bassis et al., 2021). Large areas of the West Antarctic Ice Sheet are vulnerable to this process, particularly Thwaites Glacier and Pine Island Glacier in the Amundsen Sea Sector (Scambos et al., 2017).

In response to recent climatic and oceanic change, the geographical extent of ice shelf retreat and disintegration has spread southward from the Antarctic Peninsula into West Antarctica (Cook and Vaughan, 2010; Liu et al., 2015). If the response of WAIS to a range of climate change scenarios is to be predicted with confidence, understanding the processes affecting ice shelf stability is a matter of urgency (Fox-Kemper et al., 2021). The prominent role of melt pond drainage and hydrofracturing in the demise of Larsen B in 2002 (Scambos et al., 2003) has focused attention on surface melting as a trigger for ice-shelf disintegration (e.g. Robel and Banwell, 2019; Lai et al., 2020). Recent observations, however, indicate that ice-shelf retreat and disintegration can occur in the absence of surface melt, if basal melting causes ice to lose contact with lateral or sea-floor pinning points. Examples include major rifting and calving from Pine Island Glacier following weakening of lateral pinning points (Arndt et al., 2018; Lhermitte et al., 2020) and fragmentation of the Thwaites Western Ice Tongue (TWIT; Fig. 1) following progressive loss of a sea-floor pinning point (Tinto and Bell, 2011; Miles et al., 2020).

In this paper, we focus on the Thwaites Eastern Ice Shelf (TEIS), the remaining pinned floating extension of Thwaites Glacier. TEIS is currently pinned at its distal end by a sea-floor ridge (Fig. 1), but rates of ice thinning over this pinning point suggest that complete unpinning could occur in less than one decade (Alley et. al., 2021; Wild et al. 2021). Here we use a high-frequency time series of Sentinel-1 imagery to show that within the last 5 years TEIS has transitioned from an intact ice shelf to a highly fragmented state, crossing a threshold from stable to unstable. Using the Helsinki Discrete Element Model (HiDEM), we show that this threshold-crossing behaviour was not the consequence of progressive unpinning, but occurred due to the failure of weakened ice in response to stresses associated with the pinning point. Pinning points, therefore, are not necessarily a stabilising factor for ice shelves, but instead may have a destabilising effect when ice around them is sufficiently weakened. Finally, we use the ice-sheet model BISICLES to explore the possible near-term consequences of damage evolution, shelf thinning and decoupling of TEIS from the pinning point. The recent behaviour and imminent break-up of TEIS has important implications for both ice shelf stability and the effectiveness of buttressing from intact and fragmented ice shelves.



## 2. Methods

### 2.1 Observations

We use satellite data to monitor surface features which indicate the evolution of surface and basal fractures, and to derive surface velocity fields to monitor change in ice flow rate and patterns of surface strain. The main source of satellite data is Sentinel-1 (resolution ~10m), but we also use velocity products from MODIS (Alley et al., 2021), MEaSUREs (Mouginot et al., 2017) and ITS-LIVE (Gardner et al., 2019) to provide historical context.

Surface velocity fields are derived from Sentinel-1 Interferometric Wide (IW) Mode using standard feature/speckle tracking procedures (e.g. Luckman et al., 2015). We employ the whole satellite data archive since 2014 to produce individual velocity maps on key dates and mean velocity products on an annual, quarterly or monthly basis to assess the development of speed and strain. We use a combination of 6-day and 12-day Sentinel-1 image pairs from the available archive and our feature tracking window size is 416x128 which equates to ~1km in range and azimuth. We sample the velocity field at 50 x 10 pixels before geocoding to the Antarctic Polar Stereographic projection (EPSG:3031) at 100m resolution using the REMA DEM (Howat et al., 2019) gap-filled by BedMap2 surface topography data (Fretwell et al., 2013). Strain rates are derived from selected 6-day pair velocity maps with high coherence and low noise, and are calculated in a 3x3 neighbourhood for optimum resolution.

### 2.2 Modelling

Modelling experiments were conducted with the Helsinki Discrete Element Model (HiDEM) and the BISICLES ice sheet model, to investigate fracture processes underway at TEIS and ice-sheet dynamic response to the ice shelf's evolution, respectively.

HiDEM represents ice as arrays of particles linked by breakable elastic beams, and explicitly simulates ice fracture and calving processes (Åström et al., 2013). Particles are stacked together in a hexagonal close-packed lattice to form a 3D domain representing observed ice geometries. The version used in this study (HiDEM2.0) was developed by JAÅ and Fredrik Robertsen at the CSC-IT Center for Science, Finland, with data structures and parallelisation scheme optimized for effective computation. On a modern HPC system HiDEM2.0 can compute $10^6$ timesteps for $10^8$ particles in about 24 hours, with a timestep length of 0.001 seconds. This is 1-2 orders of magnitude faster than older versions of HiDEM, allowing simulations of much larger domains with smaller particle sizes.

Five parameters determine the bulk tensile and shear strength of the ice: particle size, beam width to particle diameter ratio, beam tensile breaking strain, maximum beam endpoint bending angle, and density of randomly-distributed pre-broken beams (damage). In the simulations reported here, we use a particle size of 40m, a beam width to particle diameter ratio of 0.6, tensile breaking strain of 0.0005, and a maximum bending angle of 0.03 radians, values that were calibrated against observed fracture and calving patterns on the Greenlandic glacier Sermeq Kujalleq (Jakobshavns Isbrae). In the present study, ice strength was varied by adjusting initial damage density $d$ from 0 (no damage) to 0.6. This damage index represents a reduction in load-bearing area and thus has a similar physical meaning to damage as commonly defined (e.g. LeMaitre, 2012; Borstad et al. 2012). However, the dependence of modelled ice properties on particle and bond parameters means that values of $d$ are not directly comparable to damage variables used in other studies, including the BISICLES model discussed below. Ice damage typically increases during model runs as beams are broken in response to inter-particle stresses. Unlike the initial prescribed damage



*d*, which is macroscopically uniform and isotropic, emergent damage is localised and
anisotropic, typically taking the form of fractures and shear zones and therefore more closely
similar to real damaged ice (Åström and Benn, 2019).
The HiDEM model domain incorporates the entire area of TEIS and extends 20 to 30 km
upglacier of the grounding line. The domain is based on BedMachine v.2 (Morlighem, 2020),
which incorporates the REMA ice-surface elevation DEM (Howat et al., 2019), hydrostatic ice
thickness for the fully floating regions, and recent updates to the Thwaites Glacier bed and
adjacent seafloor (Jordan et al., 2020). The glacier bed DEM does not incorporate recent
data on the TEIS pinning point presented by Wild et al. (2021); implications of this omission
are discussed below.
The model domain was adjusted in a short surface relaxation in Elmer/Ice, to ensure that the
ice shelf base near the grounding line was at equilibrium with stresses in the ice as well as
buoyant forces. Ice viscosity and basal resistence were then estimated following the serial
inversion workflow described in Gladstone et al. (in prep) using MEaSUREs v.2 velocity data
(Rignot et al., 2011a, b; Mouginot et al., 2012, 2017). The REMA tile covering TEIS is based
on data acquired in 2013-2014 and the velocity data are from 2007 and 2009. The model
domain should thus be regarded as representing an approximation to conditions in recent
years rather than a specific snapshot in time. Subglacial friction coefficients for grounded
portions of the domain were determined assuming a linear bed friction law. Friction
coefficients were converted to SI units and rescaled ($\times 10^{-5}$) for use in HiDEM. Rescaling bed
friction coefficients is necessary to produce useful ice-displacement magnitudes, because
HiDEM simulates glacier sliding and fracture taking over timescales of seconds, whereas in
reality these processes take place over timescales of hours to years (van Dongen et al.,
73  2020).

In the form used in this study, HiDEM is a purely brittle-elastic model and does not include
viscous deformation of the ice. For simulations of short-lived calving processes it is sufficient
to allow the domain to evolve under the gravitational and buoyant forces arising from ice
geometry and water depth. Crevasses and rifts on ice shelves, however, typically propagate
on long timescales during which ice can undergo large displacements, and for such cases
some external model forcing is often desirable (e.g. Åström and Benn, 2019; Åström et al.,
2021). In the runs reported here, we applied a force at the upstream boundary of the domain
that produced a close approximation of the observed velocity structure near the grounding
line. The simulated TEIS domain contained $10^9$ particles and run over 36 hours using 2048
cores on the Mahti supercomputer at the CSC-IT Centre for Science.
BISICLES is a continuum ice flow model based on a vertically integrated stress balance
equation (Cornford et al 2020). Its treatment of fracture processes is limited to the calculation
of a scalar damage $D(x, y, t)$, which modifies the effective viscosity. $D(x, y, t)$ can be
calculated with a simple process model (Sun et al., 2016). That model, however, lacks the
skill to simulate the observations in TEIS. Instead, we estimate $D(x, y, t)$, together with basal
friction, through regularized optimizations conducted for monthly intervals through 2016 to
2020, using velocity data derived from the Sentinel-1 imagery. The model domain covers the
Amundsen Sea Embayment drainage basins including the entire Thwaites Glacier, and is
based upon the BedMachine v.2 ice thickness and bedrock elevation (Morlighem, 2020).
To assess the possible dynamic response of Thwaites Glacier to further changes on TEIS,
we conducted a set of BISICLES simulations to 2100 with varying damage, ice shelf





thickness $h(x, y, t)$ and pinning point friction. All simulations start with the basal friction calibrated to March 2016, which then evolves according to a regularized Coulomb law (Joughin et al., 2019, Zoet and Iverson, 2020). A simple set of calving criteria are also in play: ice is removed wherever $(1-D)h$ < 5 m or where ice speed exceeds $10^4$ m/yr. The damage calving criterion is similar to the full-depth crevasse calving law of Nick et al. (2010), and determines the position of the ice front rather than a calving rate. Four experiments are reported here, designed to assess the impact of different physical processes. Experiment 00 is a control, with $D(x,y)$ set to the values determined for March 2016, and basal melt rates are set such that ice shelf thickness remains constant throughout. Experiment E0 aimed to determine the impact of damage evolution in isolation. $D(x,y)$ was increased linearly within the ice shelf from 2016 to reach the value determined for March 2020, and then continued to increase at the same rate to 2026 at which point $D(x,y)$ ~ 1 across the shear zone, representing complete separation. As in 00, ice shelf thickness was held constant in time. Experiment ER includes both damage evolution and dynamic shelf-thickness evolution. In this experiment, damage was treated in the same way as E0 but the ice shelf thickness is allowed to evolve according to its velocity, without being permitted to contact the bedrock below: the grounding line may retreat but not advance. Finally, experiment UR 'unpins' TEIS, setting the ice thickness in the shear zone to zero in 2016. It allows the ice shelf to evolve in the same way as ER.

### 3. Observations

#### 3.1 Velocity structure 2015-2021
The large-scale velocity structure of TEIS underwent little change over the period 2015 to 2020 (Fig. 2). On all velocity maps, ice flow vectors on to the shelf are aligned approximately normal to the general trend of the grounding line, and over most of the shelf ice progressively veer towards the north-east as ice approaches the pinning point. The only exception to this pattern is in a relatively small region west and south-west of the pinning point, where ice flow is predominantly towards the north-west. Over most of the shelf, flow speeds are typically in the range 1 to 3 m/day and an order of magnitude lower above and adjacent to the pinning point. Sustained shrinkage of the area bounded by the 0.2 m/day velocity contour indicates progressive loss of traction over the pinning point, consistent with previous work that demonstrated ongoing ice thinning in this region (Alley et al., 2021; Wild et al. 2021).

#### 3.2 Fracture and strain 2015 - 2021
Although the large-scale velocity structure of TEIS has remained relatively constant from 2015 to 2020, considerable changes in fracture patterns and strain rates occurred during this period. In Sentinel-1 imagery, evidence for fracturing takes two forms: 1) broad surface troughs indicating depressions above basal crevasses, and 2) sharp-edged linear features indicating surface crevasses or full-depth rifts (e.g. Luckman et al., 2012). In the earliest Sentinel-1 images (2014), sets of sub-parallel basal crevasses occur down-flow of the grounding line and up-flow of the pinning point (Fig. 3). The latter set extends diagonally across the shelf from south-west to north-east, coincident with the transition between rapidly and slowly moving ice shown in Figure 2. This region is interpreted as a narrow shear zone (S: Fig. 3; Alley et al., 2021) between slowly-moving ice over the pinning point (P) and more rapidly-flowing ice on the main part of the shelf, hereafter termed the *TEIS shear zone*. In late 2016 and early 2017, a set of secondary basal crevasses started to develop at a high angle to the TEIS shear zone, on the upglacier (southern) side (T: Fig. 3). These subsequently propagated southward across the shelf and increased in number. During 2017, the easternmost of these secondary crevasses developed into a full-depth rift, visible as a sharp-





edged feature on the Sentinel imagery and leading to the detachment of a tabular berg ~18
km long and up to 2 km wide (C1, Fig. 3). Other areas of active calving are evident on both
the eastern (C2) and western (C3) flanks of TEIS, the latter releasing tabular bergs into the
area formerly occupied by the shear margin between TEIS and TWIT. Evidence for full-depth
rifting within the TEIS shear zone was initially confined to short wing cracks across blocks
bounded by basal crevasses, but became increasingly widespread through time.
Rapid evolution of the shear zone since 2018 is further illustrated by velocity gradients and
patterns of shear strain (Figs. 4 and 5). In 2018, a narrow band of high strain existed in the
northeastern portion of the shear zone, while strain in the remainder of the shear zone was
lower in magnitude and distributed over a much wider area. In late 2020, a second narrow
shear band appeared at the south-west end of the TEIS shear zone, then both bands
connected and extended across the full extent of the shelf by early 2021, indicating a
transition to full-thickness fracture. Velocity gradients across the south-western half of the
shear zone (Fig. 5) show distributed strain in the first quarter of 2020, suggesting
predominantly viscous deformation or strain distributed across numerous fractures. During
2020 all velocity profiles show increasing localization of strain as the shear zone transitioned
to full-depth fracture. Increasing shear localization is accompanied by an increase in ice
velocity on the upglacier side of the shear zone and a decrease on the downglacier (pinning
point) side. On the upglacier side, ice acceleration is substantial and ongoing (Fig. 6), with
>40% increase in ice speed in the central part of TEIS between 2018 and 2021.
**3.3 Long-term perspective: Velocity and strain from MODIS data 2002 - 2021**
The shear zone on TEIS was already a well-established feature in 2014, when the Sentinel-1
data begins. Evolution of TEIS prior to 2014, and the possible orgin of the shear zone, can be
inferred using MODIS and other historical data (Alley et al., 2021). These data are at a lower
spatial resolution (500 m) than the Sentinel-1 imagery, and strain rates were calculated over
a longer length scale (2500 m). This precludes detailed analysis, but the MODIS data provide
a valuable record of large-scale patterns of velocity and strain and their changes through
time.
Figure 6 shows a time series of velocity in the central part of TEIS, c. 15 km upglacier of the
pinning point (magenta circle on Fig. 4). The ice accelerated between 2002 and 2006, then
rapidly decreased to a minimum in 2009 before gradually increasing until 2020 when the current
quasi-exponential acceleration began. The 2002-2006 acceleration is attributed to strong
coupling between TEIS and TWIT (Alley et al., 2021). Thwaites Western Ice Tongue increased in
speed after 2002 (Miles et al., 2020), possibly in response to progressive unpinning (Tinto and
Bell, 2011), and the strong shear margin between the two portions of the shelf allowed TEIS to be
dragged forward (Alley et al., 2021). After 2006, the shear margin between TEIS and TWIT
fragmented and weakened, reducing coupling and allowing TEIS to decelerate. Meanwhile, TWIT
fragmented, transitioned into a 'mélange ice shelf' and continued to accelerate (Miles et al., 2020;
Alley et al., 2021).
Patterns of strain on TEIS for three key time periods are shown in Figure 7. During the TEIS
acceleration event (2005 - 2006), a band of positive (dextral; blue) shear strain outlines the shear
margin between TEIS and TWIT, while a zone of negative (sinistral; red) shear is evident around
the locus of the modern TEIS shear zone. Very high compressive longitudinal strain rates
upstream of the TEIS pinning point indicate significant backstress during the acceleration event.
After the acceleration event (2009-2010), the shear margin between TEIS and TWIT weakened
(Lhermitte et al., 2020; Alley et al., 2021), reducing the coupling between TEIS and TWIT, and
allowing TEIS to slow down again. This appears in the middle column of Fig. 7, where high shear





strain rates delimit the weakened shear margin between TEIS and TWIT. In contrast, both shear strain and longitudinal strain rates are very low upstream of the TEIS pinning point. However, the first large fractures within the TEIS shear zone are visible in the MODIS imagery (marked with an arrow). Data for 2019-2020 show a renewed increase of shear strain in the vicinity of the TEIS shear zone and the development of extensive rifting. This phase of shear zone evolution has been discussed in detail above (Figs. 4 & 5).

In summary, TEIS has undergone a process of fragmentation in the last five years, including brittle failure along a major shear zone up-glacier of the pinning point, formation of tensile cracks on the main body of the shelf, and release of tabular bergs on both eastern and western flanks of TEIS. This pattern of failure is consistent with longitudinal (flow-parallel) compression and transverse extension of the shelf. The origins of the TEIS shear zone can be traced at least to the 2002 - 2006 accleration event on TEIS, when zones of high longitudinal and shear strain developed upglacier of the pinning point, indicating high backstress at that time. Fragmentation of the shelf increased since 2014, with increasing shear localization and extensive full-thickness rifting since late 2020. Weakening of the shear zone has been accompanied by rapid (possibly exponential) acceleration of the ice on the upglacier side.

## 4. Modelling

### 4.1 Fracture modelling with HiDEM

Experiments with HiDEM were designed to test two hypotheses on the causes of the observed fragmentation of TEIS: 1) fragmentation is related to a reduction of backstress on the shelf, consequent upon progressive unpinning; and 2) fragmentation is related to progressive weakening of the ice and continuing backstress from the pinning point. To test these hypotheses, we conducted a matrix of runs with different combinations of friction over the pinning point and ice damage density. The results show that widespread fracturing of the shelf does not occur in runs with low damage density ($d$ < 0.3) but does occur for runs with high damage density ($d$ = 0.6). Results from two runs with $d$ = 0.6 and different pinning point friction are presented here.

Results for $d$ = 0.6 and pinning point friction coefficients rescaled from the Elmer/Ice inversion (hereafter: *baseline friction*) are shown in Figure 8. The pinning point does not exert any significant influence on the pattern of ice displacement, and ice is able to slide over the proximal side of the submarine ridge then calves at its crest. Extensive rifting and calving occurs on the eastern flank of TEIS (highlighted by discontinuities in displacement magnitudes), but only limited fracturing occurs on the western flank.

Figure 9 shows results for $d$ = 0.6 and a no slip boundary condition over the pinning point. In this case, a teardrop-shaped area of stagnant, largely intact ice extends upglacier from the pinning point. Patterns of ice displacement on TEIS show progressive veering towards the north-east as it approaches the stagnant zone. Mobile and stagnant ice are separated by a shear zone, indicated by the sharp discontinuity in displacement magnitudes. The east and west flanks of TEIS exhibit widespread propagation of subparallel rifts and calving of tabular icebergs and tensile fractures are developed in the central part of the shelf (Fig. 10). Unlike the separate linear fractures observed on TEIS, the tensile fractures are part of stepped features with both shear and tensile components. This difference probably reflect the lack of viscous processes in the model.

The fracture pattern in the 'no slip' simulation thus exhibits many similarities with that observed in February 2021 (cf. Figs. 1 & 10). Key common features are: 1) a triangular zone of slowly moving or stagnant ice extending upglacier from the pinning point; 2) deflection of ice flow around the stagnant zone; 3) development of a shear zone (TEIS shear zone) upglacier of the pinning point; 4) and rifting and calving along the eastern and western flanks of TEIS. The close similarity



between the observed pattern of fracture and the 'no slip' HiDEM simulation support the
conclusion that the recent fragmentation of TEIS occurred in response to uniaxial compression
(i.e. opposing driving stress and backstress from the pinning point) and absence of transverse
confining pressure.
**4.2 BISICLES model results: damage evolution and future dynamics**
An initial set of simulations with BISICLES was conducted to optimize basal traction and
damage $D$, using monthly velocity data for the period 2016 and 2020. Results for the month
of March are shown in Figure 11. Model speeds match observed speeds across the majority
of Thwaites Glacier: the notable exception is TWIT, where brittle facture and the formation of
a mélange shelf are poorly described by the continuum model. There the mismatch is as
much as 1.3 m/day, or 10% of the observed speed. The observed increase in speed in TEIS
between 2016 and 2020 can be reproduced in the model by minor changes in the basal
traction and by a strip of damage coincident with the shear zone adjacent to the pinning point
whose magnitude increases over time.
We then used the optimized basal traction and damage values in March 2016 as the starting
point for forward model simulations (Fig. 12). In all cases, quite large changes occur to the
velocity pattern on TEIS. Experiment 00 (control experiment with damage and ice shelf
thickness held constant at March 2016 values) exhibits a reduction in speed of ~0.5 m/day
across both ice shelf regions and immediately upstream by 2032, caused by the thinning of
the ice upstream and the resulting reduced gravitational driving stress. Experiment E0
(linearly extrapolated damage, constant ice shelf thickness) produces an increase in speed
on TEIS upglacier of the pinning point, attributed to reduced ability of the shelf to support
backstress. In contrast, ice around the grounding line of TWIT and the eastern grounding line
of TEIS decreases in speed, for similar reasons to that in the control experiment. Experiment
ER (extrapolated damage, dynamically evolving ice shelf thickness) results in faster flow over
most of TEIS, with the ice shelf acceleration leading to thinning and a further loss of
buttressing. The greatest speed increase is seen in experiment UR, because here resistive
stresses are reduced as in ER and the shelf also becomes unconfined on the western flank.
In all cases, there is at most modest increase in the ice flux across grounding line onto TEIS,
even when large acceleration occurs on the shelf. This is consistent with inversion results
that indicate that at the present time, backstress from TEIS does not make a significant
contribution to the force balance along much of the grounding line.
The future contribution of Thwaites Glacier to sea level rise computed by the BISICLES
model depends only weakly on changes in TEIS. Figure 13 shows the discharge of ice above
flotation, that is $V = \int_{\Omega_G} \vec{\nabla}_H \cdot (\vec{u}\,(h - h_f))\, d\,\Omega_G$, where $\vec{u}$ is the horizontal ice velocity, $h$ is ice
thickness, $h_f$ is the flotation thickness, $\vec{\nabla}_H$ is the horizontal gradient operator, and $\Omega_G$ is the
grounded part of the Thwaites drainage basin. Although the models with extrapolated
damage and evolving ice thickness do exhibit greater discharge over the first few years, all
four experiments show the same long-term trend, tending toward a common equilibrium. The
brief spikes in discharge evident in experiments 00 and E0 have nothing to do with the ice
shelves: instead, the artificial maintenance of ice shelf thickness the region to the west of the
ice shelves result in large ice speeds and a (modelled) calving event that reduces basal
traction in a region clearly visible in Figure 11.



## 5. Discussion

### 5.1 Observed and modelled fracturing

Sentinel-1 data, available since 2014, show that TEIS has transitioned from a largely intact ice shelf into an extensively fractured state, with full-depth fracturing along the TEIS shear zone, rifting and tabular calving along both eastern and western flanks, tensile fracturing in the central part of the shelf, and acceleration of ice flow. Concurrently, ice has been progressively decoupling from the submarine pinning point at its distal end.

The HiDEM modelling highlights the key role of backstress from the pinning point in this pattern of fracturing. The TEIS shear zone does not develop in runs where friction over the pinning point has the baseline values derived by rescaling output from the Elmer/Ice inversion. In contrast, a shear zone and other features similar to the observed fracture pattern do develop in runs where a no-slip boundary condition is imposed over the pinning point. It may appear paradoxical that the shear zone fails to develop under model boundary conditions derived from observed ice geometry and dynamics. Two reasons may be suggested for this.

First, the inversion-derived friction coefficients over the pinning point may be unrepresentative because of uncertainties associated with the sea-floor topography, the inversion process in Elmer/Ice, and the subsequent rescaling for HiDEM. Radar data show the bedrock topography is up to 200 m shallower than indicated by the gravity inversion used as the basis for the model bed (Wild et al., 2021). In the Elmer/Ice inversion, the small area of the pinning point may mean that the inverted basal resistance is too low as the result of spatial smoothing in the regularization process. Rescaling of friction to account for the different functional timescales of HiDEM and Elmer/Ice introduces further uncertainties, particularly because HiDEM does not incorporate viscous processes. Taken together, these factors could mean that the bed is 'too slippy' over the pinning point in the HiDEM simulations using the baseline friction values.

Second, the necessity for high pinning point friction to create a shear zone in HiDEM may mean that the observed TEIS shear zone was initiated at a time when pinning point friction was greater than it was when the model input data were acquired (2007 and 2009 for MEaSUREs and 2013-2014 for REMA). It is therefore possible that the TEIS shear zone developed when the pinning point provided greater backstress, and that progressive weakening through shear localisation and damage evolution has allowed the shear zone to persist although pinning point friction has been diminishing. Whatever the case, the HiDEM results are unequivocal: the observed fracture pattern requires backstress from the pinning point sufficient to initiate and sustain shear failure in the shelf. That is, backstress must consistently exceed the evolving effective strength of TEIS.

In the HiDEM runs, a high initial damage density ($d$ = 0.6) is required to produce a shear zone. This does not imply that a similar degree of pervasive damage was required to initiate the TEIS shear zone. As noted above, the properties of ice in HiDEM reflect a range of parameters and we chose damage density as a transparent and easily tunable control on ice strength. It must also be emphasised that the version of HiDEM used here is purely brittle-elastic, and does not incorporate viscous deformation. This means that the model cannot include processes likely to have been important on TEIS, particularly viscous shear localisation in the early development of the shear zone. In addition, evolution of TEIS likely involved a number of factors not included in the simulations, such as stress concentrations associated with basal roughness elements such as advected crevasses and basal channels.





## 5.2 Causes of fragmentation of TEIS

The transformation of TEIS from a largely intact ice shelf into its currently fragmented state suggests the pinning point has shifted from being a stabilising to a destabilising influence. Formerly, backstress from the pinning point was sufficient to constrain ice flow, but insufficient to initiate fracturing. At some point the balance shifted, such that backstress from the pinning point exceeded the effective strength of the ice, and damage was able to accumulate. This shift may have occurred for three fundamental reasons:

1. Changes in resistive stresses at the grounded margins of the shelf;

2. Changes in stress patterns due to interactions between TEIS and TWIT; and

3. Changes to the effective strength of the shelf.

The grounded margins of TEIS have undergone overall retreat in recent decades. The area of ice-bed contact at the TEIS pinning point has reduced (Wild et al., 2021), and the grounding line at the upglacier boundary of TEIS has retreated (e.g. Rignot et al., 2014; Milillo et al., 2019). In addition, interferograms of TEIS for the 1990s indicate the presence of a possible additional pinning point in the middle of TEIS (Rignot, 2001), which is not evident in more recent times. While reduction in the area of the TEIS pinning point may be expected to have reduced backstress on the shelf, the other changes may have increased longitudinal compressive stresses on the floating ice.

Concomitant with the changes to grounded margins, there have been large changes in the relationship between TEIS and TWIT. First, acceleration of TWIT after 2002 (itself likely a response to weakening of a sub-shelf pinning point) was transmitted across shear margin to TEIS, causing the observed speed-up between 2002 and 2006 (**Fig. 6**). This speed-up was associated with increased longitudinal and shear strain on TEIS, indicating increased loading on the shelf. Subsequently, fragmentation and opening of the shear margin exposed the western flank of TEIS, reducing lateral confining stress on that side.

Several factors may have contributed to weakening TEIS in recent decades. Basal melting in response to incursions of warm, deep water has reduced ice-shelf thickness, with basal melt rates typically on the order of 5 m/yr (Seroussi et al. 2017; Wåhlin et al., 2021; Alley et al., 2021). Focused basal melting in sub-shelf channels can increase basal roughness, create local stress concentrations and interact with fracture processes. Weakening may also occur through the advection of damage from upglacier, and basal crevasses or other fractures could seed additional damage when they reach different stress regimes. Finally, damage can accumulate through time via positive feedbacks between damage and strain (Åström and Benn, 2019; Lhermitte et al., 2020).

The recent evolution of TEIS may have involved a combination of all the above factors. On the balance of available evidence, we propose the following sequence of events as the most likely cause of the recent fragmentation. The TEIS shear zone was initiated, or was significantly modified, in the mid-2000s, when TEIS accelerated in response to stresses transferred across the strong shear margin with TWIT (Alley et al. 2021). During the acceleration event, high longitudinal and shear strain rates developed on TEIS in response to elevated longitudinal compression supported by backstress from the pinning point. Large fractures within the TEIS shear zone first appear in satellite imagery after the acceleration event, suggesting that damage was initiated or increased in response to elevated stresses



(Fig. 7). Fragmentation, opening and significant weakening of the shear margin between
TEIS and TWIT removed confining pressure from the western flank of TEIS, encouraging
transverse extension in response to ongoing longitudinal compression. Concurrently, damage
on the shelf continued to accumulate due to positive feedbacks between damage and strain,
and basal melting reduced ice-shelf thickness and possibly contributed to weakening the TEIS
shear zone (Wåhlin et al., 2021). Sentinel-1 data show increasing development of tensile
fractures since 2017 in response to transverse extension (Fig. 3), and increasing
concentration of strain and extensive rifting within the TEIS shear zone since 2020 (Fig. 5).
Thus, although backstress from the pinning point has likely diminished through time (Wild et
al., 2021), accumulation of damage on TEIS has ensured that the ice in the shear zone has
remained the weakest link in the system. Fragmentation of TEIS was not a consequence of
unpinning; on the contrary, fragmentation reflects stresses originating at the pinning point
acting on progressively weakening ice.
Backstress from the pinning point, once a crucial source of support for TEIS, is now hastening its
destruction. Complete fragmentation of TEIS appears to be imminent and disintegration could
occur sooner than the ~10 year timescale implied by rates of thinning and unpinning (cf. Wild et
al., 2021). Complete loss of the shelf may follow, although the former pinning point and fast ice
may retard the evacuation of icebergs in much the same way as currently observed on TWIT.

## 5.3 Future implications of TEIS loss

The BISICLES model results indicate that loss of strength in the shear zone or unpinning of
TEIS will likely have little impact on basin-wide discharge from Thwaites Glacier. There is
currently no evidence that the imminent loss of TEIS will hasten marine ice sheet instability or
the demise of Thwaites Glacier. However, future evolution of Thwaites may be significantly
influenced by calving processes along unbuttressed grounding lines, and it is possible that
calving and iceberg overturn may increase, perhaps substantially, if the ice shelves are lost.
Water depths along the grounding line of TEIS are currently in the range 400 - 700 m, which is
below the likely threshold for ice-cliff instability (cf. Bassis and Walker, 2012; Crawford et al.,
2020). The glacier bed deepens substantially 40 km upglacier of the current grounding line and
Thwaites Glacier may enter the MICI regime if the ice front retreats to that point. Until then, ice
retreat will involve calving by processes such as melt-undercutting, longitudinal extension and
buoyant calving (e.g. Benn et al., 2007; Benn and Åström, 2019). Detailed observations and
process modelling will be required to understand how ice shelf removal will affect calving
processes and ice-retreat rates in this region (e.g. Winberry et al., 2020; Crawford et al., 2020).

## 5.4 The destabilising role of pinning points in ice shelf loss

Observations of Antarctic ice shelf disintegration have implicated two main mechanisms for ice
shelf collapse: hydrofracture and unpinning from bedrock features (e.g. Doake et al., 1998;
Scambos et al., 2003). In the context of ice-shelf collapse, hydrofracturing requires a combination
of surface meltwater and stressed ice. By offsetting the effects of lithostatic pressure, water in
surface crevasses can shift an ice shelf into a tensile stress regime, leading to deep penetration
of the crevasses and runaway failure (Scambos et al., 2003; Robel and Banwell, 2019; Lai et
al., 2020). Supraglacial stream networks can transport water off ice shelves or focus it in
vulnerable areas, and hence play either a stabilising or destabilising role in the hydrofracture
process (Kingslake et al., 2017; Bell et al., 2017; Dell et al., 2020). Alternatively, unpinning
from ice shelf margins or grounding points impacts ice shelf stability by removing sources of
resistive stress. Unpinning can occur in response to increased basal melting, accumulation of
damage, or other factors including reduction of landfast sea ice (Doake et al., 1998; Glasser and
Scambos, 2008; Lhermitte et al., 2020). Some ice-shelf break-up events involve a combination of





unpinning and hydrofracture (e.g. Larsen B: Doake et al., 1998; Scambos et al., 2003), while
other cases indicate that unpinning can lead to ice-shelf fragmentation in the absence of surface
melt (e.g. TWIT: Miles et al., 2019).
Our results indicate that a third mechanism, failure of ice triggered by backstress from a pinning
point, can also lead to ice-shelf fragmentation. We propose the term *backstress-triggered failure*
for this mechanism. The contrast between backstress-triggered failure and unpinning is
instructive. Pinning points are sources of resistive stress that oppose gravitational driving
stresses. In structurally intact ice shelves, the opposition of resistive stress and driving stress is
manifest as compressive membrane stresses, including across-flow oriented 'compressive
arches' and upflow oriented backstress (Doake et al., 1998; van der Veen, 2013). Removal or
weakening of a pinning point (complete or partial unpinning) breaks the symmetry between
resistive and driving stress, removes or reduces the associated compressive membrane
stresses, and renders the ice vulnerable to *failure in tension*. In contrast, during backstress-
triggered failure the resistive stress - driving stress opposition remains in place, but instead of
supporting membrane stresses the ice *fails in compression*. The fundamental difference between
the unpinning and backstress-triggered failure scenarios is the location of the weakest link in the
system: in the former it is the contact between the shelf and the pinning point, in the latter it is the
ice itself. Pinning points and promontories at the margins of ice shelves are known to serve as
nucleation points for fractures (e.g. Arndt et al., 2018; De Rydt et al., 2019), but pinning points
generally act as stabilizing features (e.g. Favier et al., 2016; Still et al., 2019). On TEIS, the
switch from pinning-point enabled stability to backstress-triggered failure occurred as a
consequence of a number of factors, including interactions between TEIS and TWIT, removal of
lateral confining pressure, shelf thinning from basal melting, and positive feedbacks between
damage and strain. Similar factors may also prove a lethal combination on other ice shelves.
On ice shelves in steady state, pinning points play a crucial role in their stability and in
buttressing the upstream flow of ice off the Antarctic continent. If increasing basal melt and/or
changing flow dynamics upset this balance, stresses are likely to concentrate upstream of
pinning points, causing irreversible damage. In most circumstances, pinning points are essential
for ice shelf stability, but as ice shelves thin and weaken, pinning points can plant the seeds of
their demise.

**Code and data availability**

Data sources are cited in the text. Code for HiDEM, Elmer/Ice and BISICLES is open-source and
freely available.

**Author contributions**

DB and AL designed and oversaw the study in discussion with JB and EP; remote sensing
analyses were conducted by AL (Sentinel-1), SB (DEMs) and KA (MODIS); computer modelling
was done by JÅ and AC (HiDEM), RG, AC and TZ (Elmer/Ice) and SC (BISICLES). DB wrote the
manuscript with input from all authors.

**Competing interests**

The authors declare that they have no conflict of interest.

**Acknowledgements**

This work is from the DOMINOS and TARSAN projects, components of the International



Thwaites Glacier Collaboration (ITGC). Funding was provided by the National Science
Foundation (NSF: Grants 1738896 and 1929991) and Natural Environment Research Council
(NERC: Grant NE/S006605/1). Rupert Gladstone is supported by Academy of Finland grant
number 322430, Thomas Zwinger by grant number 322978. Logistics provided by NSF-U.S.
Antarctic Program and NERC-British Antarctic Survey. Sentinel-1 data were provided by the
Copernicus Program of the European Commission. ITGC contribution number ITGC-XXX.

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

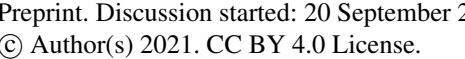

# Figures

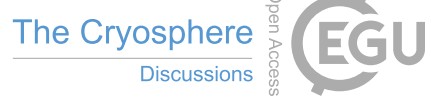

**Figure 1**: The floating extensions of Thwaites Glacier, showing location of the Eastern Ice Shelf (TEIS) the Western Ice Tongue (TWIT) and the position of grounding lines and the location of pinning points in 2011 from Rignot et al. (2014). Image based on modified Copernicus Sentinel-1 data (2014-2021), SAR-processed by ESA.

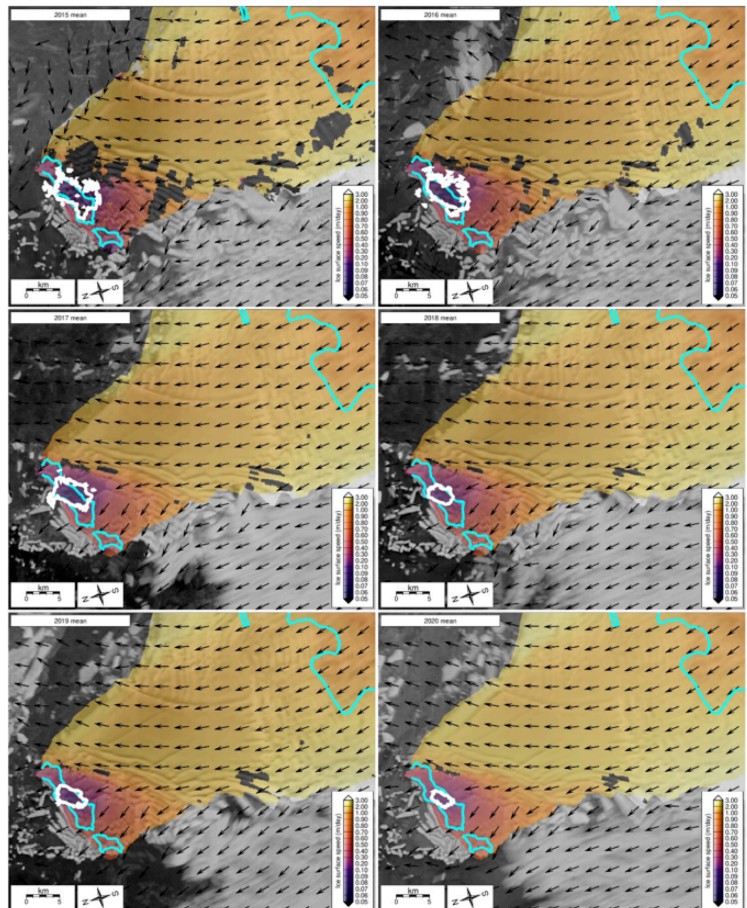

**Figure 2**: Time series of mean annual velocity on TEIS derived from Sentinel-1 speckle-
tracking between 2015 and 2020. Arrows show flow direction. Cyan lines are MEaSUREs
InSAR-derived grounding lines from 2011 (Rignot et al., 2014). White lines delineate the 0.2
m/day velocity contour which is centred on the pinning point. Images contain modified
Copernicus Sentinel-1 data (2014-2021), SAR-processed by ESA.



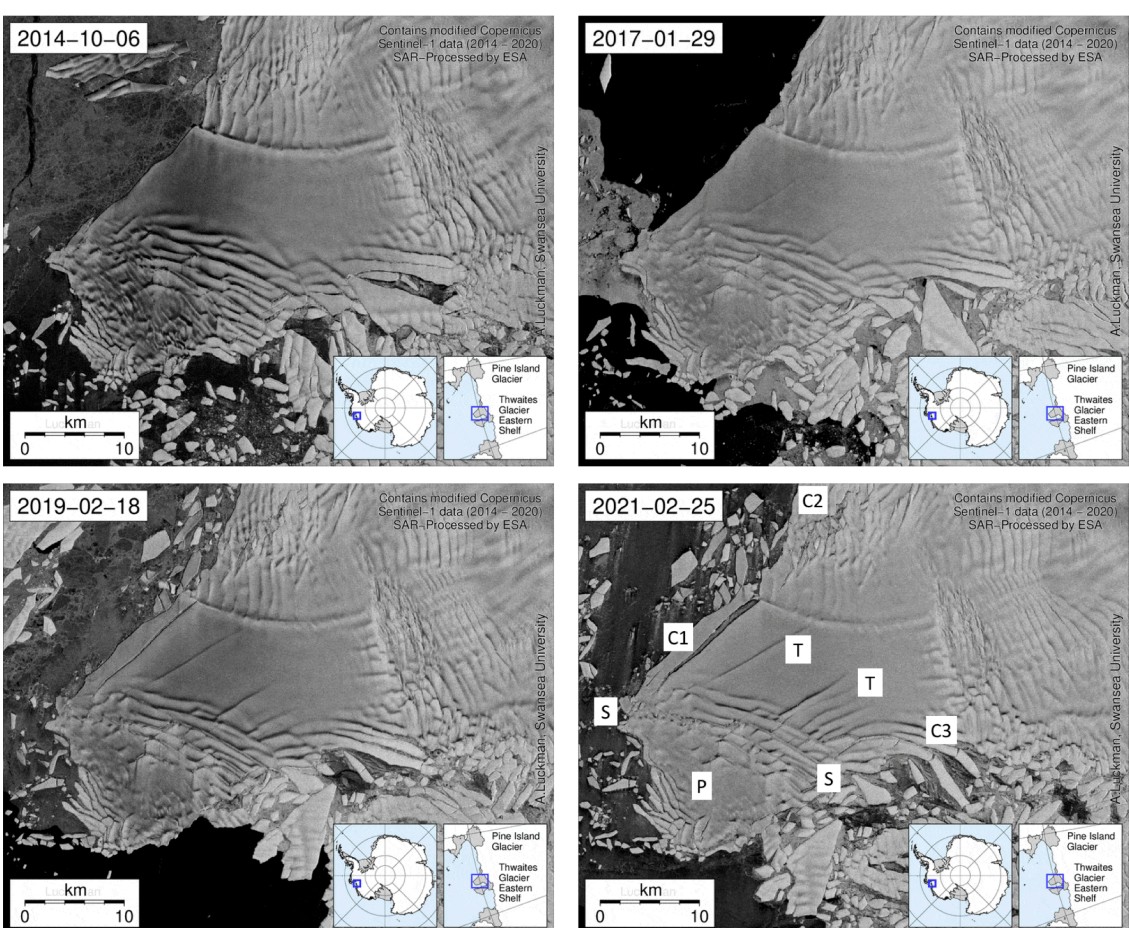

**Figure 3:** Fracture patterns on Thwaites Eastern Ice Shelf (TEIS) 2014 - 2021. The final frame labels key features described in the text: S - S: TEIS shear zone; T: tensile cracks; C1 - C3: calving along rifts; P: slow-moving, relatively intact ice above pinning point. Images contain modified Copernicus Sentinel-1 data (2014-2021), SAR-processed by ESA.

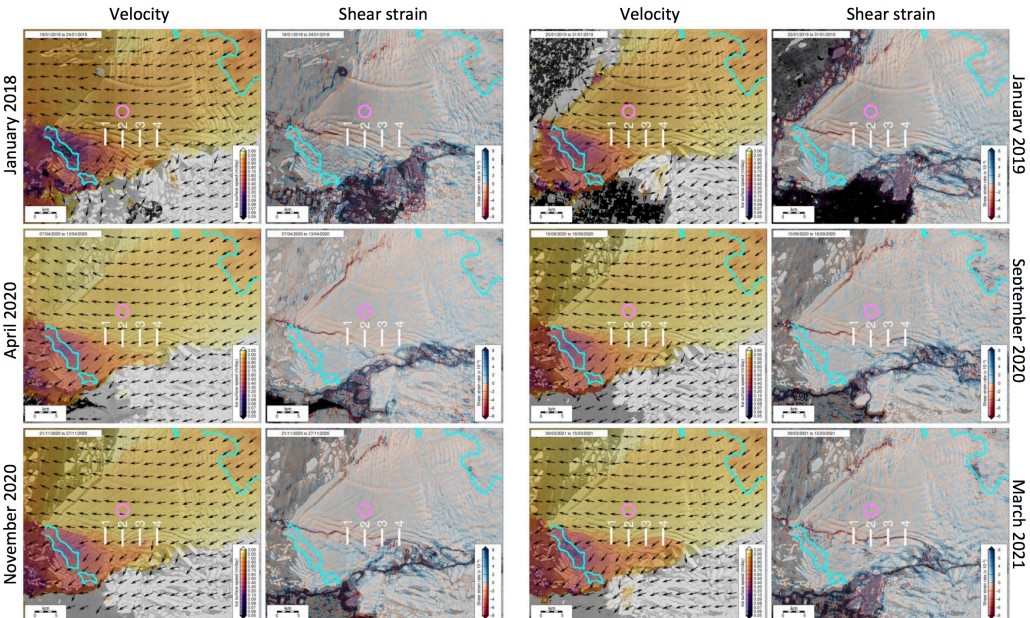

**Figure 4:** Evolution of velocity (first and third columns) and shear strain (second and fourth columns) on TEIS derived from Sentinel-1 speckle-tracking between 2016 and 2021. Specific image pairs are chosen for excellent coherence and minimal noise, and to focus on recent months. Cyan lines are MEaSUREs InSAR-derived grounding lines from 2011 showing the location of historic known pinning points. White lines show positions of numbered profiles used to extract velocities presented in **Figure 5**. The magenta circle indicates the location of the velocity time series shown in **Figure 6**. Images contain modified Copernicus Sentinel-1 data (2014-2021), SAR-processed by ESA.





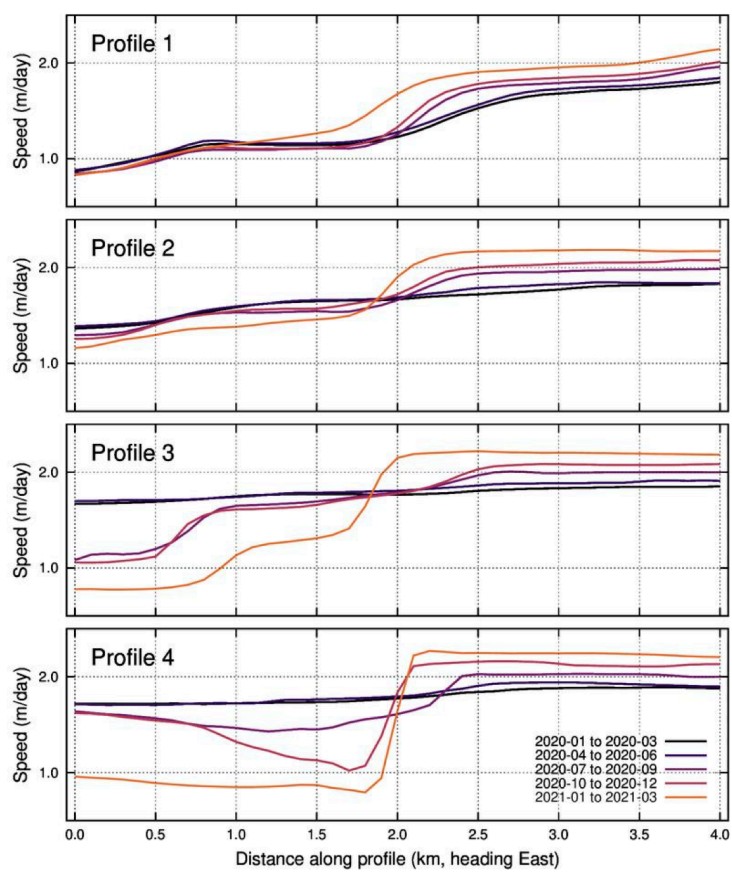

**Figure 5**: Profiles of mean quarterly surface velocity during 2020 and 2021 from Sentinel-1 speckle/feature tracking along the white lines shown in **Fig. 4**.

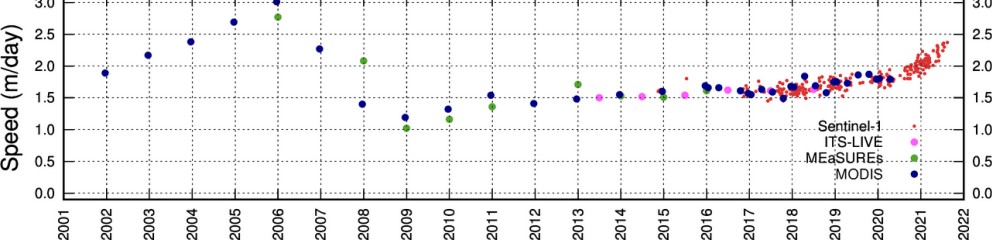

**Figure 6:** Velocity evolution at a point in the centre of TEIS from MODIS (blue dots; Alley et al., 2021), ITS_LIVE (pink), MEaSUREs (green) and Sentinel-1 (red) showing the 2002-2006 acceleration, the 2006-2008 slowdown, 2008-2020 modest acceleration and recent rapid acceleration as the shelf crossed the fracture transition. Location of velocity data is shown by the magenta circle in **Figure 4**.



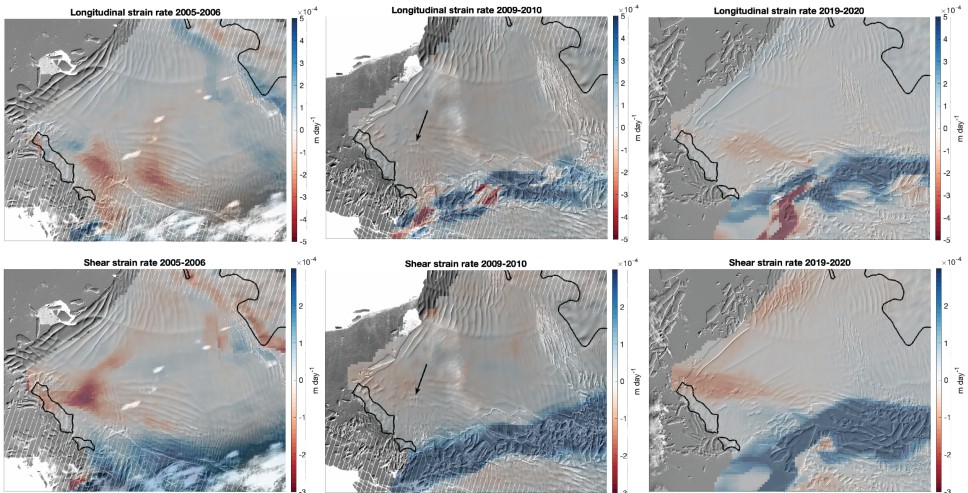

**Figure 7**: Mean longitudinal strain rates (top row) and shear strain rates (bottom row), in
2005-2006 during the TEIS acceleration event (left-hand column), 2009-2010 (middle
column), and 2019-2020 (right-hand column). The TEIS-TWIT shear margin is near the
bottom of each image and the TEIS shear zone is to the right of the twin 'islands' of the
pinning point, outlined in black. Crevasses in the location of the current shear zone are
indicated by the arrow in the middle column. Landsat-7 (first and second column) and
Landsat-8 (third column) imagery courtesy of the U.S. Geological Survey. Strain rates are
from Alley et al. (2021).

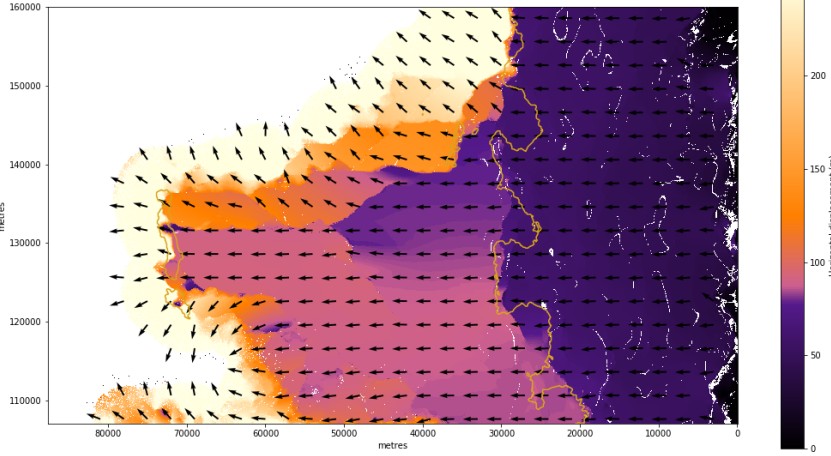

**Figure 8**: Ice displacement pattern simulated in HiDEM model run with the baseline friction
boundary condition over the pinning point (i.e. values derived from the Elmer/Ice inversion



rescaled for HiDEM) and damage density = 0.6. Grounding lines are indicated by orange
lines. The observed shear zone (cf. **Fig. 3**) does not develop in this simulation. The zone of
large displacements around TEIS (yellow) consists of calved icebergs.

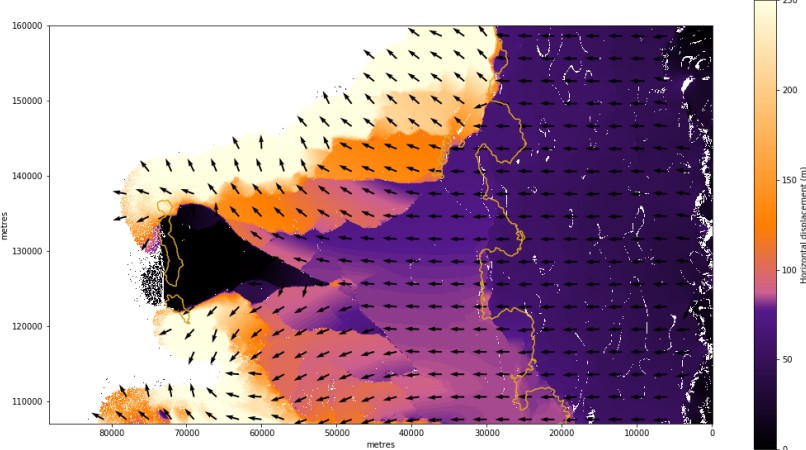

**Figure 9**: Pattern of ice displacement simulated in HiDEM model run with a no-slip boundary
condition over the pinning point and damage density = 0.6. Note the triangular area of
stationary ice extending upglacier from the pinning point, and the sharp displacement
gradient (shear zone) between it and the main body of TEIS.

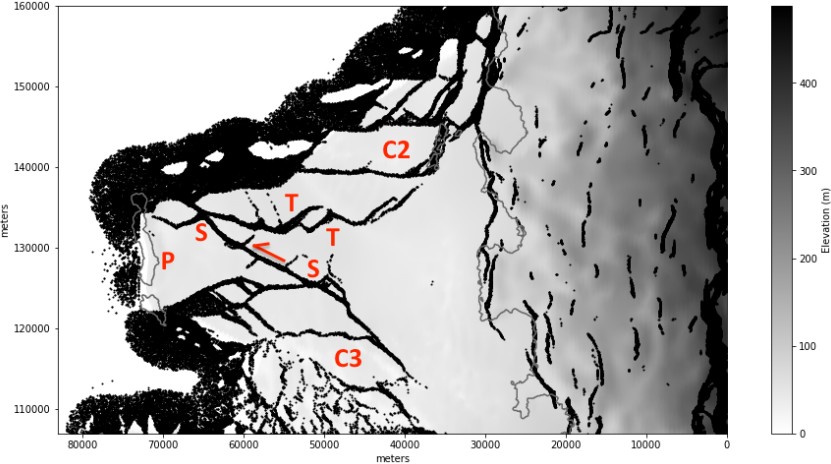

**Figure 10:** Fracture pattern simulated in HiDEM for the no-slip boundary condition over the
pinning point and damage density = 0.6. Bonds broken during the simulation are indicated in
black, surface elevation in greyscale shading, and grounded areas delineated by grey lines.
Letters indicate features analogous to the observed features in **Fig. 3**; P: pinning point; S:
shear zone; C2, C3: rifting and calving on the eastern and western flanks; and T: tensile
cracks.



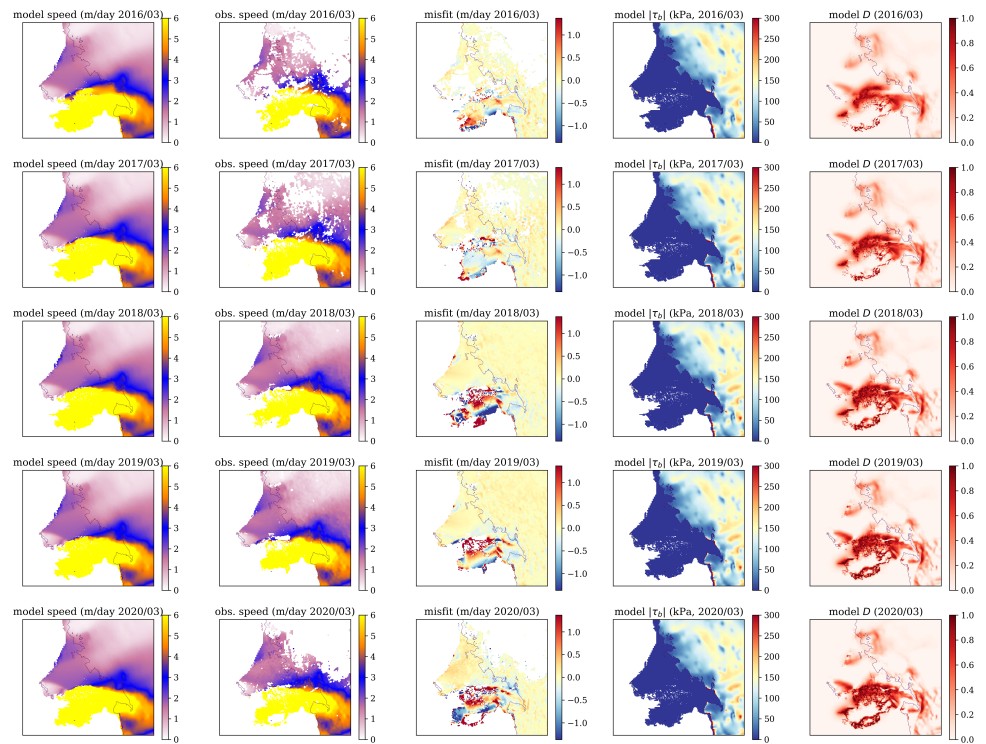

**Figure 11**: BISICLES inverse problem results, March 2016-2020. The first and second columns show the optimized model velocities and observed velocities; the third column shows the misfit bewteen modelled and observed velocities; and the fourth and fifth columns show the resulting optimized basal traction and damage. Each row shows data for March of one year.



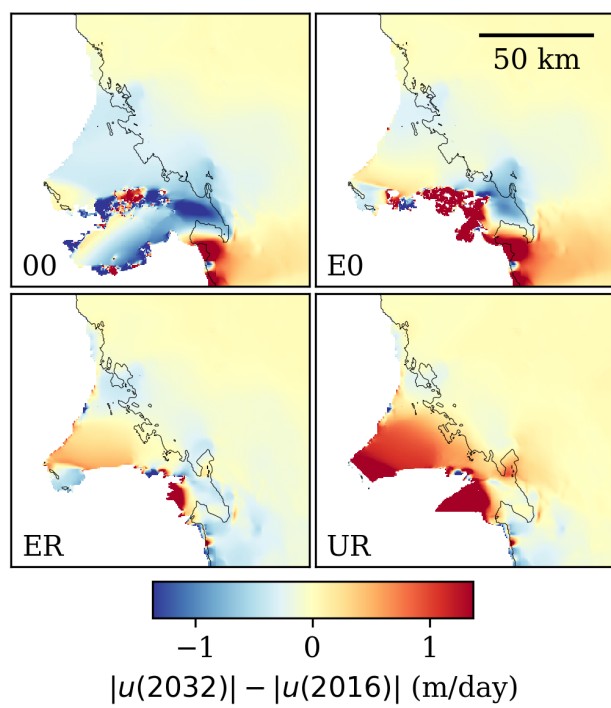

**Figure 12**: Output of BISICLES simulations of TEIS, showing differences in velocity relative to 2016. For details of each run, see text.

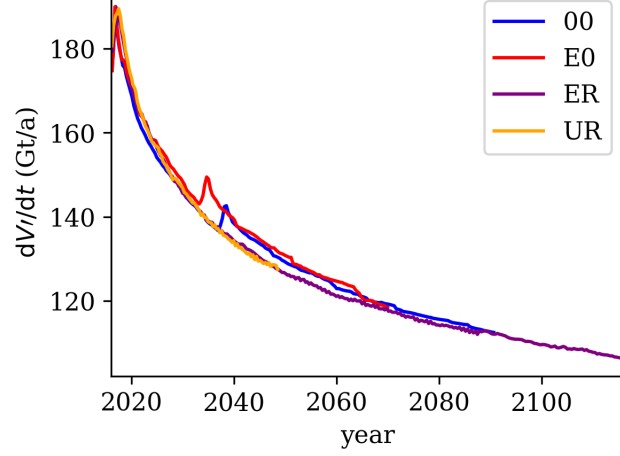

**Figure 13**: Discharge of ice above flotation $V$ associated with modelled changes to TEIS.