# Peer review of "Rapid fragmentation of Thwaites Eastern Ice Shelf, West Antarctica"

_The Cryosphere, 2021_

## Referee Comment (RC1)

**Review of tc-2021-288**

November 9, 2021

**General comments**

Benn et al. present a comprehensive analysis of the processes that have contributed to the weakening and fragmentation of the Thwaites Eastern Ice Shelf (TEIS). They begin with a detailed description of the recent changes in ice velocity, strain rates and fracture patterns inferred from Sentinel-1 imagery, with a particular focus on the progressive weakening and development of the shear zone upstream of the TEIS pinning point. The discrete element model HiDEM is used to simulate fracture development under two conditions: low basal friction versus a 'no slip' boundary condition over the pinning point. Similarities between the modelled and observed fracture patterns lead the authors to conclude that relatively high backstress from the TEIS pinning point is responsible for the extensively fractured ice-shelf state. Additional prognostic experiments performed with the ice-sheet model BISICLES show that ungrounding of the TEIS pinning point or additional ice-shelf damage will not significantly increase mass loss from the Thwaites Glacier basin. Altogether, the authors demonstrate that the TEIS pinning point currently acts as a destablising feature because the pinning point backstress is sufficient to trigger the failure of unconfined, damaged ice undergoing thinning.

This manuscript is timely and of scientific interest given the projected rapid retreat and mass loss from the Amundsen Sea glaciers. Overall, the manuscript is well-written, enjoyable to read, and it provides a valuable record (and explanation) of the processes leading to the destabilisation of TEIS. The use of two different modelling approaches involving an elastic fracture model and a continuum ice dynamics model, combined with the detailed, high temporal resolution analysis of recent Sentinel-1 imagery, is where the manuscript builds on previous analyses of the weakening of TEIS. My main concerns are with the assumptions made about pinning point basal friction, and the need for additional detail about the model representation of the pinning point.

**Specific comments**

- The conclusion that high pinning point backstress is responsible for the pattern of failure across TEIS (Pg. 9, L9) required a 'no-slip' boundary condition over the model pinning point, resulting in a fairly large zone of zero-displacement upstream. Could this be an overestimation of the basal friction provided by the pinning point? In reality, the pinning point does not reduce ice velocity to zero (Fig. 4 suggests flow speeds of 0.1 to 0.5 m per day over the pinning point). After finding that the inferred pinning point friction coefficient from the Elmer/Ice inversion was too low to modify the pattern of ice displacement, why not increase the friction coefficient over the pinning point area (since you are already rescaling the friction coefficient anyway for Hi-DEM). This could be an alternative to using a more extreme (and somewhat unrealistic) no-slip boundary condition over the pinning point that appears to overestimate the backstress provided as shown by the large area of stationary ice in Fig. 9. Similarly, in the discussion, you justify the requirement for a high damage density of 0.6 in order to produce a shear zone, but you haven't justified the requirement for very high basal drag provided by the HiDEM pinning point.

- The statement that there is close similarity between the observed Feb 2021 pattern of fracture and the 'no-slip' simulation (Pg. 8, L51) would be more convincing if Fig. 10 and Fig. 3d (2021) were presented beside each other. Fig. 10 has a different coordinate system and orientation to Figs. 3 and 1 (the pinning point has rotated by 45 degrees in Fig. 10), and as result, it is not easy to pick similarities between the two (even with the labels). Including the pinning point outline in Fig. 3 may also help. This comparison is important given that one of the main conclusions from the HiDEM 'no-slip' experiment is that recent fracturing and TEIS fragmentation is due to the backstress provided by the pinning point, rather than gradual ungrounding and a reduction in backstress.

- Pg. 2, L73: The citations provided are not examples of ice shelf disintegration occurring in response to loss of contact with pinning points.

- Pg. 2, L86: I don't think it is correct to say that TEIS crossed a threshold from stable to unstable within the last 5 years when there is much evidence to suggest that TEIS was undergoing gradual change prior to 2016. This also depends on whether you define an unstable ice shelf state as undergoing irreversible change, or by some other definition.

- Pg. 4, L57: You mention that the DEM doesn't include recent data from Wild et al. (2021) on the TEIS pinning point, but it would be useful to provide more information on how the pinning point is actually represented in the model geometry. Does the model pinning point consist of two separate pinning points or one broader grounded region? (You have to zoom quite far in to Fig. 11 to see this). What is the model pinning point height above flotation and is it comparable to the different height above flotation calculations for the same pinning point by Wild et al. (2021)? What is the difference between the modelled and observed ice velocity over the pinning point? Since BISICLES simulations are conducted to show that removal of the pinning point will have no influence on ice loss, you should demonstrate that care has been taken ensure the accurate representation of both model pinning point morphology, and flow resistance provided by the pinning point.

- Pg. 4, L60: What was the time period required to relax the model, and did the model relaxation change the geometry near the TEIS pinning point? Is the model pinning point area and height above flotation still representative of the realworld pinning point after relaxation?

- Pg. 5, L9&15: At this stage of the paper, it isn't clear whether you are referring to the shear zone immediately upstream of the pinning point, or the shear margin between TEIS and TWIT. The TEIS shear zone is introduced in the following section.

- Pg. 5, L15: How large is the area where the ice thickness is set to zero to simulate unpinning? This could also be indicated in a figure.

- Pg. 5, L97-98: Did you vary the pinning point friction, or remove the pinning point entirely?

- Pg. 5, L98: Why did you choose not to relax the model before each simulation?

- Pg. 5, L99: How did the friction coefficient pattern evolve during each forward experiment in comparison to the 2016 basal friction? Did the friction coefficient over the pinning point also evolve in Experiments 00, E0 and ER?

- Pg. 8, L87: Fig. 13 shows that the discharge of ice above flotation, $V$, decreases by approximately 30% by 2100 in each BISICLES simulation. This is not intuitive and the reasons for this decrease deserve some further discussion.

- Pg. 8, L96: It's not clear where this region of reduced traction is in Fig. 11.

- Fig 12: Is there a reason why you chose to use year 2032 to compare to year 2016? 16 years doesn't seem a sufficient amount of time to allow a model to adjust to a perturbation such as unpinning or an increase in damage. Do the speed changes shown Fig. 12 persist after the year 2032 or is there a further change in speeds as the model readjusts to a new steady state?

- Fig 13: Why does the line for experiment UR end at 2050, experiment E0 end at 2070, and experiment ER end at year 2120 if you ran each simulation until 2100 as stated in the method? As shown, the figure doesn't support the claim that all of the experiments show the same long-term trend if the change in $V$ until 2100 isn't shown for each of the four simulations.

- Pg. 9, L10: Why not modify the model seafloor topography by $+200$ m in order to achieve a more accurate height above flotation at the pinning point location? The BISICLES simulations demonstrate that unpinning will have very little impact on ice discharge from Thwaites Glacier, but if the bathymetry is too deep, is it possible that you are underestimating the flow-resisting effect of the pinning point?

- Pg. 11, L41: Neither of these studies implicate unpinning as a mechanism of ice shelf collapse.

**Technical corrections**

- Pg. 3, L6: Provide the resolution of the other three velocity products, similar to the Sentinel-1 description.

- Pg. 2, L16: BedMap2 = Bedmap2

- Pg. 3, L26: Begin the paragraph with: "HiDEM is a brittle-elastic fracture model that can be used to simulate…" And then continue with the explanation of how ice is represented as arrays of particles.

- Add north arrows to Figs. 2 and 3. In the text you refer to the regions southwest and northeast of the pinning point.

- The manuscript has two subsections entitled 'Modelling'. The paper would be easier to follow if you changed the first to 'Model experiments' or the second to 'Model results'

- Figs. 2, 4, 7. The resolution is too poor and the text size too small to read the text by the colourbar. Alternatively, use one larger colourbar corresponding to all of the subplots.

- Fig. 4. The legend says shear strain rate, but the unit suggests strain.

- Fig. 6. Do the different dot sizes represent the velocity error or something else?

- Fig. 9. Why does the pinning point outline extend beyond the no-slip region? Is the pinning point grounding line in this figure the modelled grounding line from Elmer/Ice after relaxation?

- Fig. 12. Is the grounding line in the figure the model grounding line at year 2032?

---

## Referee Comment (RC2)

Review

**Rapid fragmentation of Thwaites Eastern Ice Shelf, West-Antarctica**

Summary

Benn and colleagues present a set of observations (velocity changes, fracture observations, strain) to document the weakening and fragmentation of the Thwaites Eastern Ice Shelf (TEIS) and combine that with two models (HiDEM and BISICLES) to assess the role of the submarine pinning point on that fragmentation. By comparison of HiDEM with the observations, they conclude that the observations best match the 'no slip' scenario, leading to the conclusions that the high backstress conditions from the pinning point causes the observed fragmentation. Secondly, they use different BISICLES scenarios to assess the role of further damage, thinning or unpinning on ice sheet discharge and come the the conclusion that further fragmentation or unpinning does not have a large influence on mass loss from the Thwaites basin.

**General remarks**

The paper is well written, touches an important and timely topic and provides several new insights on the role of pinning points on ice shelf stability. Therefore, I think the paper can be recommended for publication in TC, given that the authors address the detailed comments raised below. Most of these comments relate to rewording statements, weakening some claims and/or adjusting figures to make them publication-proof.

**Detailed comments**

P1 - L43: "*backstress triggered failure*" I am personally not 100% convinced this failure mechanism can be considered a third mechanism that can be compared on an equal level to hydrofracturing and unpinning. The failure observed here is a combination of factors (as also highlighted in the discussion) and these are I think the drivers/mechanism, not the backstress as such. The backstress basically stays equal and the rest changes, so I believe the different changes are the drivers/mechanism and not the backstress as such.

P2 - L66: "*southward*"; it is perhaps a semantic comment wind directions are always relative in Antarctica. In my humble opinion, moving from the Peninsula to West Antarctica corresponds mostly to going westward and not to going southward.

P2 - L86-87 "*threshold-crossing behaviour*": the observed changes are indeed important etc. but I do not see any analyses/proof of threshold-crossing behaviour. What is the threshold for considering an ice shelf fragmented? One rift? Multiple rifts? Where is the threshold between stable/unstable? I think the observed changes correspond to gradual changes and I do not see any analysis on the paper of where the threshold is. Having such a quantitative threshold would be meaningful for future studies but is not part of the paper.

P2 - L87-89 "*we show that this threshold-crossing behaviour was not the consequence of progressive unpinning, but occurred due to the failure of weakened ice in response to stresses associated with the pinning point*" I am not convinced that the experiments completely support this claim. The HiDEM scenarios are two (unrealistic) extremes and I agree that it corresponds better to the no-slip condition, but this not necessarily mean that it is not the consequence of progressive unpinning as the reality might be somewhere in between.

P3 - L 13: how are the different velocity data combined? Do the authors account for double counting some observations in the 6-12 day pairs?

P3 – L16: Which REMA product? Mosaic or strips?

P3 – L41-42 *"that were calibrated against observed fracture and calving patterns on the 142 Greenlandic glacier Sermeq Kujalleq (Jakobshavns Isbrae)"* Based on which study? Reference?

P4 - L64 *"REMA tile"* which tile? Why tiles and not the mosaic? Could be clarified with better description of which data is being used (see also earlier comment).

Section 2.2: I miss a clear overview of the experiments being performed. E.g. there is no description of the HiDEM experiments (this is postponed to section 4.1), whereas there is a (difficult to follow) description of the BISICLES experiments. I would be very helpful for the reader to have a complete, uniform overview for the different HiDEM/BISICLES experiments (and their motivation) in section 2.2. Adding a table with the experiment settings would also increase readability/interpretability.

P4 – L89-90 "That model lacks the skill" I guess the standalone model without inversion? If so, please clarify that, because it not clear where "that model" refers to. Additionally, what is meant by "*lacks the skill*"? Clarify.

P4 – L97 "*simulation to 2100*" Not all time series in Fig 13 go to 2100. Some go further and others stop earlier.

P5 – L27: "*above*" description of directions can be misleading. Is that above in figure direction or above in stream direction? Please use consequently directions relative to flow directions.

Section 3.2 and Figure 3: I did find it not easy to see the described features in the panels. I would be helpful to indicate that on the respective panels and not only in the last panel.

P6 – L67+79 "*upglacier*" Not sure if this is correct English.

P7 L20-27: should be part of the method section and the settings for HiDEM for these runs should be better explained (time period, friction, pinning points, etc).

P7 – L30: "*baseline friction*" This implies in my opinion that there is friction and does not correspond to the earlier statement of "*progressive unpinning*".

Figure 8-10: it would be helpful for comparison if Fig-10 would be merged showing:
- on left panel: baseline friction condition of Fig. 8 (with fractures like in Fig.10 superimposed)
- on right panel: no slip condition of Fig. 9 (with fractures like in Fig.10 superimposed)

P8 - L59: I wonder what the added value of Fig 11 is. It is not really used in the paper, except to show that BISICLES makes sense. Could be moved to the SM.

P8 - L88 Equations should be added as equations and not as part of the text.

Fig. 13 + Section 5.3 is counterintuitive and differs in my opinion strongly from state-of-the-art. Why would the discharge in ice above flotation decrease with time to half of the initial values? This counters moreover the work of Hongju et. al. (https://tc.copernicus.org/articles/12/3861/2018/) which show a constant or increasing discharge of ice above flotation. This should be discussed in much more detail

This counterintuitive decrease, moreover, raises doubt about the validity of the claim that further damaging and/or unpinning will not have any significant impact on the future ice discharge. This would also imply the current buttressing effect of the ice shelf is negligible, which would surprise me.

Conclusion: I miss a conclusion section where the results are repeated and summarized.

Figure 2+4: Color bars, legends etc cannot be read as they are too small. It would be beneficial for the readability of each figure to have one common colorbar for similar panels.

Figure 3: figure seems gathering of individual figures (from other sources like twitter ;-) ) and should be adapted from distracting features to be publication ready. Suggestions:
- remove *Sentinel-1 ESA* statements
- remove *Luckman, Swansea University* statements
- remove Antarctica, study area subpanels as they are part of Figure 1 already.

Fig.8-9. Color bar should be updated to scientific colorbar which is more homogeneous. Now there are sharp color contrasts for some differences (e.g. between 70m (purple), 80m (pink) and 90m (orange)) whereas they are gradual for others (100-160m is all orange). There is no reason why the 20m difference between 70-90m is more important than 100-120m so the color panels should also be continuous. I suggest reading https://www.nature.com/articles/s41467-020-19160-7 for proper colorbar selection.

Fig.10. fractures in black have the same color as maximum elevation, which makes interpretation difficult. Please use different color for fractures or elevation.

Fig.13 why do each of the simulations have a different end date? Does that matter?

---

## Author Comment (AC1)

**Comments and **Responses**, CC1: Community Comment, Dustin Schroeder**

This is an exciting paper about an important ice shelf.  This is just a comment to share that Figure 10 in our paper (https://www.pnas.org/content/116/38/18867) includes radar sounding images of ice shelf thinning and crevasse formation from 1979 and 2009 which may provide useful context for the observations and processes presented in your paper.

**RESPONSE: Thanks Dustin. It is useful to have brought this to the attention of readers, but we don't think it requires any changes to the manuscript.**

---

## Author Comment (AC2)

**Comments and **Responses**, RC1: Anonymous Referee #1**

**GENERAL COMMENTS**

Benn et al. present a comprehensive analysis of the processes that have contributed to the weakening and fragmentation of the Thwaites Eastern Ice Shelf (TEIS). They begin with a detailed description of the recent changes in ice velocity, strain rates and fracture patterns inferred from Sentinel-1 imagery, with a particular focus on the progressive weakening and development of the shear zone upstream of the TEIS pinning point. The discrete element model HiDEM is used to simulate fracture development under two conditions: low basal friction versus a 'no slip' boundary condition over the pinning point. Similarities between the modelled and observed fracture patterns lead the authors to conclude that relatively high backstress from the TEIS pinning point is responsible for the extensively fractured ice-shelf state. Additional prognostic experiments performed with the ice-sheet model BISICLES show that ungrounding of the TEIS pinning point or additional ice-shelf damage will not significantly increase mass loss from the Thwaites Glacier basin. Altogether, the authors demonstrate that the TEIS pinning point currently acts as a destablising feature because the pinning point backstress is sufficient to trigger the failure of unconfined, damaged ice undergoing thinning.

This manuscript is timely and of scientific interest given the projected rapid retreat and mass loss from the Amundsen Sea glaciers. Overall, the manuscript is well-written, enjoyable to read, and it provides a valuable record (and explanation) of the processes leading to the destabilisation of TEIS. The use of two different modelling approaches involving an elastic fracture model and a continuum ice dynamics model, combined with the detailed, high temporal resolution analysis of recent Sentinel-1 imagery, is where the manuscript builds on previous analyses of the weakening of TEIS. My main concerns are with the assumptions made about pinning point basal friction, and the need for additional detail about the model representation of the pinning point.

**RESPONSE: We are very grateful to the reviewer for their summary and helpful comments.**

**SPECIFIC COMMENTS**

• The conclusion that high pinning point backstress is responsible for the pattern of failure across TEIS (Pg. 9, L9) required a 'no-slip' boundary condition over the model pinning point, resulting in a fairly large zone of zero displacement upstream. Could this be an overestimation of the basal friction provided by the pinning point? In reality, the pinning point does not reduce ice velocity to zero (Fig. 4 suggests flow speeds of 0.1 to 0.5 m per day over the pinning point). After finding that the inferred pinning point friction coefficient from the Elmer/Ice inversion was too low to modify the pattern of ice displacement, why not increase the friction coefficient over the pinning point area (since you are already rescaling the friction coefficient anyway for Hi-DEM). This could be an alternative to using a more extreme (and somewhat unrealistic) no-slip boundary condition over the pinning point that appears to overestimate the backstress provided as shown by the large area of stationary ice in Fig. 9. Similarly, in the discussion, you justify the requirement for a high damage density of 0.6 in order to produce a shear zone, but you haven't justified the requirement for very high basal drag provided by the HiDEM pinning point.

**RESPONSE: HiDEM simulates the elastic component of the TEIS deformation. The no-slip boundary condition used in HiDEM is considered appropriate over the short-duration time scales of motion considered.** *We will add a few sentences in the text to address this topic.*

• The statement that there is close similarity between the observed Feb 2021 pattern of fracture and the 'no-slip' simulation (Pg. 8, L51) would be more convincing if Fig. 10 and Fig. 3d (2021) were presented beside each other. Fig. 10 has a different coordinate system and orientation to Figs. 3 and 1 (the pinning point has rotated by 45 degrees in Fig. 10), and as result, it is not easy to pick similarities between the two (even with the labels). Including the pinning point outline in Fig. 3 may also help. This comparison is important given that one of the main conclusions from the HiDEM 'no-slip' experiment is that recent fracturing and TEIS fragmentation is due to the backstress provided by the pinning point, rather than gradual ungrounding and a reduction in backstress.

**RESPONSE: Great idea -** *We will include Figures 8, 9 and 10 as a multi-panel figure, with the TEIS model output re-oriented to match the orientation of the TEIS in earlier figures. Figure 3d will be included as the*

*fourth panel and the labelling that was originally included on Figure 3d will be removed. We will also include the pinning point outline in Figure 3.*

• Pg. 2, L73: The citations provided are not examples of ice shelf disintegration occurring in response to loss of contact with pinning points.

**RESPONSE: There are no citations in L73. The citations on L72 are in support of a different mechanism for ice shelf disintegration (melt and ponding) and this passage is simply setting the scene.**

• Pg. 2, L86: I don't think it is correct to say that TEIS crossed a threshold from stable to unstable within the last 5 years when there is much evidence to suggest that TEIS was undergoing gradual change prior to 2016. This also depends on whether you define an unstable ice shelf state as undergoing irreversible change, or by some other definition.

**RESPONSE: We stand by the language used. In Figure 5 we present profiles of velocity that show a distinct transition from a largely intact to a fully broken shear margin. Figure 6 shows the velocity response of crossing this threshold, which occurs in early 2020. More recent data (which we will update Figure 6 with) show further acceleration of the eastern part of the shelf.** *Nevertheless, we will carefully review our use of the term 'threshold' in response to this comment.*

• Pg. 4, L57: You mention that the DEM doesn't include recent data from Wild et al. (2021) on the TEIS pinning point, but it would be useful to provide more information on how the pinning point is actually represented in the model geometry. Does the model pinning point consist of two separate pinning points or one broader grounded region? (You have to zoom quite far in to Fig. 11 to see this). What is the model pinning point height above flotation and is it comparable to the different height above flotation calculations for the same pinning point by Wild et al. (2021)? What is the difference between the modelled and observed ice velocity over the pinning point? Since BISICLES simulations are conducted to show that removal of the pinning point will have no influence on ice loss, you should demonstrate that care has been taken ensure the accurate representation of both model pinning point morphology, and flow resistance provided by the pinning point.

**RESPONSE: As in reality, the model pinning region consists of two separate pinning points. The height above floatation at the pinning point is ~ 20 m with a small local maximum of 31 m. These values are comparable to those shown by Wild et al. (2021) and any differences are negligible considering the 40 m particle size used in our HiDEM simulations. The surface velocity following inversion is in agreement with observed velocities over the grounded region (~150 m a$^{-1}$ or less).** *We will update the text at this point in the paper to include these points so as to satisfy readers similarly interested in the details.*

• Pg. 4, L60: What was the time period required to relax the model, and did the model relaxation change the geometry near the TEIS pinning point? Is the model pinning point area and height above flotation still representative of the real world pinning point after relaxation?

**RESPONSE: The domain was relaxed for a short period of 2.5 days, over which minimal change occurred to the TEIS geometry over the pinning point. The extent of the pinned ice, obtained from BedMachine (v2020), was unchanged over the relaxation period. The surface height over the pinned ice changed between 0 and 2 m, with a small number of locations in the distal region of the TEIS were modified by up to 4 m. The minimal surface height change over the pinning point during relaxation, and the domain's similarity to that shown in Wild et al. (2021), means that the TEIS domain is representative of the real-world pinning point.** *To address this comment, we will provide more information on the model relaxation and limited change to model geometry in the text.*

• Pg. 5, L9&15: At this stage of the paper, it isn't clear whether you are referring to the shear zone immediately upstream of the pinning point, or the shear margin between TEIS and TWIT. The TEIS shear zone is introduced in the following section.

**RESPONSE: We were referring to the shear margin between TEIS and the pinning point.** *This will be clarified in the text.*

• Pg. 5, L15: How large is the area where the ice thickness is set to zero to simulate unpinning? This could also be indicated in a figure.

RESPONSE: **We set thickness to zero over an area of floating ice surrounding the pinning point.** *We will add a figure showing the region in question.*

• Pg. 5, L97-98: Did you vary the pinning point friction, or remove the pinning point entirely?

RESPONSE: **'pinning point friction' was not the best choice of words here; we meant something like 'the buttressing imposed on the ice shelf by the pinning point', but this is covered by the two previous items (damage, ice thickness). To clarify, w**e will remove 'and pinning point friction' to make this sentence clear. However, the overall evolution of friction across the region is important and we have carried out additional simulations to quantify that, which will be included in improved figures.*

• Pg. 5, L98: Why did you choose not to relax the model before each simulation?

RESPONSE: **There is in fact a brief relaxation which is integrated with the optimization for friction and damage coefficients.** *We will clarify this in the text.*

• Pg. 5, L99: How did the friction coefficient pattern evolve during each forward experiment in comparison to the 2016 basal friction? Did the friction coefficient over the pinning point also evolve in Experiments 00, E0 and ER?

RESPONSE: **The friction coefficient does not evolve, but the friction does. The basal friction in all forward runs is given by a simple rule that depends on two parameters, a field $\beta(x, y)$ (the basal friction coefficient), which is estimated in the optimization process, and a regularization speed, $u_0$.** *We will clarify this in the text.*

• Pg. 8, L87: Fig. 13 shows that the discharge of ice above flotation, V, decreases by approximately 30% by 2100 in each BISICLES simulation. This is not intuitive and the reasons for this decrease deserve some further discussion.

RESPONSE: **Both reviewers commented on this. It occurs because our experiments simulate the reduction of the pinning point influence while the rest of the ice shelf remains close to present day conditions. The result is an initial acceleration followed by a gradual deceleration as the systems tends to a new equilibrium dependent on (for example) the buttressing provided by the ice shelf in the region of the present day grounding line. This is distinct from typical TG simulations (e.g. Hongju et al. (2018) as one reviewer mentions) that apply a melt rate sufficient to ablate the ice shelf substantially and prevent substantial future ice shelf formation.** *To allow the reader to relate our results to typical simulations, we have carried out additional simulations, with a melt rate taken from Hongju et al. (2018). In these simulations we see that same sort of results as Hongju et al. (2018), i.e. sustained mass loss at rates at and above the present day, with some variation between them due to the unpinning. These additional simulations will be explained and presented.*

• Pg. 8, L96: It's not clear where this region of reduced traction is in Fig. 11.

RESPONSE: *We will improve Fig. 11 to make this clearer.*

• Fig 12: Is there a reason why you chose to use year 2032 to compare to year 2016? 16 years doesn't seem a sufficient amount of time to allow a model to adjust to a perturbation such as unpinning or an increase in damage. Do the speed changes shown Fig. 12 persist after the year 2032 or is there a further change in speeds as the model readjusts to a new steady state?

RESPONSE: **The ice flow reacts immediately to some perturbations (e.g. the unpinning of U0) and over a longer period to others (e.g. the increased flow results in the shelf thinning and in turn some loss of buttressing upstream). The year 2032 was chosen to represent a medium term response, because it shows some key differences (e.g. that most simulations have not slowed in the same way as the control (00)). To address the reviewer's concerns, w**e will add the results for some more years to an appendix.*

• Fig 13: Why does the line for experiment UR end at 2050, experiment E0 end at 2070, and experiment ER end at year 2120 if you ran each simulation until 2100 as stated in the method? As shown, the figure doesn't

support the claim that all of the experiments show the same long-term trend if the change in V until 2100 isn't shown for each of the four simulations.

**RESPONSE: This was an omission on our part. Since most of the simulations showed the same behaviour, we did not run all to 2100.** *We have now run all of them to 2116 (i.e., for 100 years) and include the updated result in the figures.*

• Pg. 9, L10: Why not modify the model seafloor topography by +200 m in order to achieve a more accurate height above flotation at the pinning point location? The BISICLES simulations demonstrate that unpinning will have very little impact on ice discharge from Thwaites Glacier, but if the bathymetry is too deep, is it possible that you are underestimating the flow-resisting effect of the pinning point?

**RESPONSE: We mention above that the height above floatation is entirely reasonable when compared to Wild et al.. However, some small local differences between our bed and the seafloor topography used by Wild et al. may still exist. Furthermore, we note for the BISICLES simulations that height above flotation at the pinning point is not as important as it might seem. What matters (to our model, where basal friction does not depend on thickness above flotation) is that the model velocity and velocity gradients surrounding the pinning point are correct. This ensures that the stress imposed in the rest of the shelf is correct, at least initially. This can be achieved (by optimizing the damage around the pinning point and the basal traction upon it) providing that the grounded region has a suitable extent. Cornford et al. (2015) did have to raise the bathymetry, but that was because the BedMap2 bedrock elevation in the region was too deep to provide any grounded region at all.** *We will update the text to clarify these points.*

• Pg. 11, L41: Neither of these studies implicate unpinning as a mechanism of ice shelf collapse.

**RESPONSE: *We will insert an alternative reference to substantiate this point.***

**TECHNICAL CORRECTIONS**

• Pg. 3, L6: Provide the resolution of the other three velocity products, similar to the Sentinel-1 description.

**RESPONSE: *Will be done***

• Pg. 2, L16: BedMap2 = Bedmap2

**RESPONSE: *Will be done***

• Pg. 3, L26: Begin the paragraph with: "HiDEM is a brittle-elastic fracture model that can be used to simulate. . . " And then continue with the explanation of how ice is represented as arrays of particles.

**RESPONSE: *We will edit this line to include the reviewer's suggested introduction to HiDEM.***

• Add north arrows to Figs. 2 and 3. In the text you refer to the regions southwest and northeast of the pinning point.

**RESPONSE: *Figure 1 provides the geographical context and Figure 2 already includes north arrows.***

• The manuscript has two subsections entitled 'Modelling'. The paper would be easier to follow if you changed the first to 'Model experiments' or the second to 'Model results'

**RESPONSE: *We will differentiate the two modelling sections by renaming the first to "Model experiments" and the second to "Model results", both naming suggestions coming from the reviewer.***

• Figs. 2, 4, 7. The resolution is too poor and the text size too small to read the text by the colourbar. Alternatively, use one larger colourbar corresponding to all of the subplots.

**RESPONSE: *Will be done***

• Fig. 4. The legend says shear strain rate, but the unit suggests strain.

**RESPONSE: *The values are strain rate and the units will be corrected.***

• Fig. 6. Do the different dot sizes represent the velocity error or something else?

**RESPONSE: The different sizes of dot are simply there to make the figure easier to understand. If the dense Sentinel-1 points were larger they would over-print each other. If the other dots representing other sources of data were smaller, they would be difficult to make out.**

• Fig. 9. Why does the pinning point outline extend beyond the no-slip region? Is the pinning point grounding line in this figure the modelled grounding line from Elmer/Ice after relaxation?

**RESPONSE: Yes, this is the modelled grounding line from the Elmer/Ice relaxation. No-slip conditions are imposed for particles interacting with the bed within the regions delineated by the grounding line. The figure is, however, illustrating surface movement, and some surface movement does occur at the ends of the elongated grounding region.**

• Fig. 12. Is the grounding line in the figure the model grounding line at year 2032?

**RESPONSE: Yes, this is the 2032 grounding line. *We will state this in the caption.***

---

## Author Comment (AC3)

**Comments and **Responses**, RC2: Referee #2: Stef Lhermitte**

**SUMMARY**

Benn and colleagues present a set of observations (velocity changes, fracture observations, strain) to document the weakening and fragmentation of the Thwaites Eastern Ice Shelf (TEIS) and combine that with two models (HiDEM and BISICLES) to assess the role of the submarine pinning point on that fragmentation. By comparison of HiDEM with the observations, they conclude that the observations best match the 'no slip' scenario, leading to the conclusions that the high backstress conditions from the pinning point causes the observed fragmentation. Secondly, they use different BISICLES scenarios to assess the role of further damage, thinning or unpinning on ice sheet discharge and come the conclusion that further fragmentation or unpinning does not have a large influence on mass loss from the Thwaites basin.

**RESPONSE: We are very grateful for the summary.**

**GENERAL REMARKS**

The paper is well written, touches an important and timely topic and provides several new insights on the role of pinning points on ice shelf stability. Therefore, I think the paper can be recommended for publication in TC, given that the authors address the detailed comments raised below. Most of these comments relate to rewording statements, weakening some claims and/or adjusting figures to make them publication-proof.

**RESPONSE: We are very grateful for the kind comments.**

**SPECIFIC COMMENTS**

P1 - L43: "*backstress triggered failure*" I am personally not 100% convinced this failure mechanism can be considered a third mechanism that can be compared on an equal level to hydrofracturing and unpinning. The failure observed here is a combination of factors (as also highlighted in the discussion) and these are I think the drivers/mechanism, not the backstress as such. The backstress basically stays equal and the rest changes, so I believe the different changes are the drivers/mechanism and not the backstress as such.

**RESPONSE: We maintain (and the reviewer does not seem to disagree) that the highlighted mechanism for ice shelf failure is distinct from hydrofracturing and unpinning, and that is a key aspect of this paper. Nevertheless, he makes a fair point that the backstress is not the only factor. We will modify the language in the abstract to downplay the impression that a single factor is at play.**

P2 - L66: "*southward*"; it is perhaps a semantic comment wind directions are always relative in Antarctica. In my humble opinion, moving from the Peninsula to West Antarctica corresponds mostly to going westward and not to going southward.

**RESPONSE: *We will modify the language to address this point.***

P2 - L86-87 "*threshold-crossing behaviour*": the observed changes are indeed important etc. but I do not see any analyses/proof of threshold-crossing behaviour. What is the threshold for considering an ice shelf fragmented? One rift? Multiple rifts? Where is the threshold between stable/unstable? I think the observed changes correspond to gradual changes and I do not see any analysis on the paper of where the threshold is. Having such a quantitative threshold would be meaningful for future studies but is not part of the paper.

**RESPONSE: Figure 6 shows a change from stable to accelerating behaviour around the beginning of 2020; this trend has continued and Figure 6 will be updated with more recent data. We maintain that this discontinuity in behaviour is consistent with a threshold being crossed. This hypothetical threshold is a bulk property of the shelf and need not be quantitative to have meaning. *Nevertheless, we will tone down the language in relations to thresholds to reflect the imprecise nature of this change.***

P2 - L87-89 "*we show that this threshold-crossing behaviour was not the consequence of progressive unpinning, but occurred due to the failure of weakened ice in response to stresses associated with the pinning point*" I am not convinced that the experiments completely support this claim. The HiDEM scenarios are two (unrealistic) extremes and I agree that it corresponds better to the no-slip condition, but this not necessarily mean that it is not the consequence of progressive unpinning as the reality might be somewhere in between.

**RESPONSE: Figure 5 shows the "failure of weakened ice" in detail. Further, it shows that ice on the pinned side of the shear margin slows down as the rift crosses the failure threshold. If pinning point weakening was important, this ice would be expected to speed-up. We maintain that this fully explains the behaviour with no need for weakening of the pinning point to be involved.** *We will make these points clearer in the text and discuss the relative importance of pinning point weakening.*

P3 - L 13: how are the different velocity data combined? Do the authors account for double counting some observations in the 6-12 day pairs?

**RESPONSE: The Sentinel-1 velocity observations are combined in different ways as explained on the preceding lines for different purposes (presented in Figures 2, 4, 5 and 6). Captions for the figures give more details about which combinations are used. Where velocity maps are averaged, we make no allowance for the fact that 12-days pairs coincide with two 6-day pairs as this would make no significant difference to the results.** *What is missing from the explanation is how noise filtering allows individual velocity maps to be combined and we will correct this omission.*

P3 – L16: Which REMA product? Mosaic or strips?

**RESPONSE: Mosaic.** *Will be clarified in the text.*

P3 – L41-42 *"that were calibrated against observed fracture and calving patterns on the 142 Greenlandic glacier Sermeq Kujalleq (Jakobshavns Isbrae)"* Based on which study? Reference?

**RESPONSE: These calibrations are not yet published. They were conducted to calibrate the model for use in multiple investigations of failure dynamics at Thwaites Glacier. Jakobshaven Isbrae was used for the calibration because of the glacier's large ice thickness and the availability of remotely-sensed observations of calving.**

P4 - L64 *"REMA tile"* which tile? Why tiles and not the mosaic? Could be clarified with better description of which data is being used (see also earlier comment).

**RESPONSE: The mosaic is delivered as a set of tiles. We refer to the tile covering TEIS because it allows us to be specific about the date.** *We will review all of the text around REMA to make sure it is all clear.*

Section 2.2: I miss a clear overview of the experiments being performed. E.g. there is no description of the HiDEM experiments (this is postponed to section 4.1), whereas there is a (difficult to follow) description of the BISICLES experiments. I would be very helpful for the reader to have a complete, uniform overview for the different HiDEM/BISICLES experiments (and their motivation) in section 2.2. Adding a table with the experiment settings would also increase readability/interpretability.

**RESPONSE:** *We will improve the overview of experiments by moving the description of HiDEM experiments to this section and clarifying the description of BISICLES experiments. We will add a table that summarizes the experiment settings for HiDEM and BISICLES, as suggested.*

P4 – L89-90 "That model lacks the skill" I guess the standalone model without inversion? If so, please clarify that, because it not clear where "that model" refers to. Additionally, what is meant by "*lacks the skill*"? Clarify.

**RESPONSE:** *We will clarify what is meant by the model lacking skill to simulate the acute fracture dynamics at TEIS. The model of Sun et al. (2016) attempts to evolve damage according to a simple relationship with local stress and damage advected from upstream, which proves too simplistic for to model TEIS and TWIT accurately.*

P4 – L97 *"simulation to 2100"* Not all time series in Fig 13 go to 2100. Some go further and others stop earlier.

**RESPONSE:** *We will ensure that all simulations run to 2116 (i.e. for 100 years).*

P5 – L27: *"above"* description of directions can be misleading. Is that above in figure direction or above in stream direction? Please use consequently directions relative to flow directions.

**RESPONSE:** *We will readily make this change.*

Section 3.2 and Figure 3: I did find it not easy to see the described features in the panels. I would be helpful to indicate that on the respective panels and not only in the last panel.

**RESPONSE: *We will readily make this change.***

P6 – L67+79 "*upglacier*" Not sure if this is correct English.

**RESPONSE: This term is used frequently (at least on this side of the pond!) and seems readily understandable to us. If the editorial team thinks it needs changing, we will find an alternative.**

P7 L20-27: should be part of the method section and the settings for HiDEM for these runs should be better explained (time period, friction, pinning points, etc).

**RESPONSE: *This paragraph will be moved to Section 2.2 as mentioned in a previous response. We will provide further detail regarding HiDEM settings and will summarize the experiments run in a table.***

P7 – L30: "*baseline friction*" This implies in my opinion that there is friction and does not correspond to the earlier statement of "*progressive unpinning*".

**RESPONSE: We are using the term to mean something different – i.e. the first value from which we make changes. *We will clarify the text to prevent this misunderstanding.***

Figure 8-10: it would be helpful for comparison if Fig-10 would be merged showing:
- on left panel: baseline friction condition of Fig. 8 (with fractures like in Fig.10 superimposed)
- on right panel: no slip condition of Fig. 9 (with fractures like in Fig.10 superimposed)

**RESPONSE: We agree with the reviewer that an alteration is needed to ease the comparison of the model outputs. *We will combine the original figures 8, 9 and 10 into a single, multi-panel figure. We decided to include the fracture plot directly under the corresponding displacement plot to ease comparison between fracture and particle displacement magnitudes. It was decided to not overlay the fractures on the displacement plots because of the congestion in the amount of information being presented.***

P8 - L59: I wonder what the added value of Fig 11 is. It is not really used in the paper, except to show that BISICLES makes sense. Could be moved to the SM.

**RESPONSE: The purpose of Fig. 11 is to show that the observed speed up in TEIS can be explained by the introduction of a specific pattern of damage. This is what we mean by "The observed increase in speed in TEIS 363 between 2016 and 2020 can be reproduced in the model by minor changes in the basal traction and by a strip of damage coincident with the shear zone adjacent to the pinning point whose magnitude increases over time". *We will rephrase that first sentence of that paragraph to make this more explicit.***

P8 - L88 Equations should be added as equations and not as part of the text.

**RESPONSE: *We will make edits so that the equation is included as an equation on its own line of text.***

Fig. 13 + Section 5.3 is counterintuitive and differs in my opinion strongly from state-of-the-art. Why would the discharge in ice above flotation decrease with time to half of the initial values? This counters moreover the work of Hongju et. al. (https://tc.copernicus.org/articles/12/3861/2018/) which show a constant or increasing discharge of ice above flotation. This should be discussed in much more detail. This counterintuitive decrease, moreover, raises doubt about the validity of the claim that further damaging and/or unpinning will not have any significant impact on the future ice discharge. This would also imply the current buttressing effect of the ice shelf is negligible, which would surprise me.

**RESPONSE: Both reviewers commented on this. It occurs because our experiments simulate the reduction of the pinning point influence while the rest of the ice shelf remains close to present day conditions. The result is an initial acceleration followed by a gradual deceleration as the systems tends to a new equilibrium dependent on (for example) the buttressing provided by the ice shelf in the region of the present day grounding line. This is distinct from typical TG simulations (e.g. Hongju et al. (2018), as one reviewer mentions) that apply a melt rate sufficient to ablate the ice shelf substantially and prevent substantial future ice shelf formation. *To allow the reader to relate our results to typical simulations, we have carried out and will present additional simulations, with a melt rate taken from Hongju et al. (2018).***

*In these simulations we see that same sort of results as Hongju et al. (2018), i.e. sustained mass loss at rates at and above the present day, with some variation between them due to the unpinning.*

Conclusion: I miss a conclusion section where the results are repeated and summarized.

**RESPONSE*: We will add a conclusion section in which we will summarize our main results.***

Figure 2+4: Color bars, legends etc cannot be read as they are too small. It would be beneficial for the readability of each figure to have one common colorbar for similar panels.

**RESPONSE:** *Will do.*

Figure 3: figure seems gathering of individual figures (from other sources like twitter ;-) ) and should be adapted from distracting features to be publication ready. Suggestions:
- remove *Sentinel-1 ESA* statements
- remove *Luckman, Swansea University* statements
- remove Antarctica, study area subpanels as they are part of Figure 1 already.

**RESPONSE: Apologies for the hasty figure!** *All these aspects will be corrected.*

Fig.8-9. Color bar should be updated to scientific colorbar which is more homogeneous. Now there are sharp color contrasts for some differences (e.g. between 70m (purple), 80m (pink) and 90m (orange)) whereas they are gradual for others (100-160m is all orange). There is no reason why the 20m difference between 70-90m is more important than 100-120m so the color panels should also be continuous. I suggest reading https://www.nature.com/articles/s41467-020-19160-7 for proper colorbar selection.

**RESPONSE:** *Figures 8 and 9 will be included in the multi-panel figure with Figure 10. The colour bar will use a continuous colour scale.*

Fig.10. fractures in black have the same color as maximum elevation, which makes interpretation difficult. Please use different color for fractures or elevation.

**RESPONSE:** *Will do.*

Fig.13 why do each of the simulations have a different end date? Does that matter?

**RESPONSE: It doesn't matter a great deal (the trends are clearly the same in each case), but we have nonetheless made sure now that all simulations end on the same date.**